# Structural basis of HIV-1 Vpu-mediated BST2 antagonism via hijacking of the clathrin adaptor protein complex 1

Xiaofei Jia[1†], Erin Weber[1†], Andrey Tokarev[2,3], Mary Lewinski[2,3], Maryan Rizk[2,3], Marissa Suarez[2,3], John Guatelli[2,3]*, Yong Xiong[1]*

[1]Department of Molecular Biophysics and Biochemistry, Yale University, New Haven, United States; [2]Department of Medicine, University of California San Diego, La Jolla, United States; [3]The VA San Diego Healthcare System, San Diego, United States

**Abstract** BST2/tetherin, an antiviral restriction factor, inhibits the release of enveloped viruses from the cell surface. Human immunodeficiency virus-1 (HIV-1) antagonizes BST2 through viral protein u (Vpu), which downregulates BST2 from the cell surface. We report the crystal structure of a protein complex containing Vpu and BST2 cytoplasmic domains and the core of the clathrin adaptor protein complex 1 (AP1). This, together with our biochemical and functional validations, reveals how Vpu hijacks the AP1-dependent membrane trafficking pathways to mistraffick BST2. Vpu mimics a canonical acidic dileucine-sorting motif to bind AP1 in the cytosol, while simultaneously interacting with BST2 in the membrane. These interactions enable Vpu to build on an intrinsic interaction between BST2 and AP1, presumably causing the observed retention of BST2 in juxtanuclear endosomes and stimulating its degradation in lysosomes. The ability of Vpu to hijack AP-dependent trafficking pathways suggests a potential common theme for Vpu-mediated downregulation of host proteins.

*For correspondence: jguatelli@ucsd.edu (JG); yong.xiong@yale.edu (YX)

†These authors contributed equally to this work

## Introduction

The interferon-inducible restriction factor BST2 (also named tetherin, CD317 and HM1.24) presents a potent innate immune restriction to many enveloped viruses (*Neil et al., 2008*; *Van Damme et al., 2008*; *Jouvenet et al., 2009*; *Evans et al., 2010*; *Arias et al., 2012*). BST2 has a short cytoplasmic tail at the N-terminus followed by a single transmembrane (TM) helix, a long coiled-coil extracellular domain, and a C-terminal glycosyl phosphatidylinositol (GPI) anchor (*Kupzig et al., 2003*). The availability of two membrane anchors connected by a long coiled-coil allows BST2 to efficiently inhibit viral transmission by tethering newly formed virions at the infected cell surface and preventing their release to the surrounding environment (*Hinz et al., 2010*; *Schubert et al., 2010*; *Yang et al., 2010*). The short intracellular, N-terminal tail of BST2 has been implicated in the natural trafficking of this antiviral protein (*Rollason et al., 2007*; *Masuyama et al., 2009*).

HIV-1 overcomes restriction by BST2 using the viral protein Vpu (*Neil et al., 2008*; *Van Damme et al., 2008*). The mechanism by which Vpu antagonizes BST2 appears to be multifaceted, involving both degradation and mistrafficking within the endosomal system. Vpu (about 81 amino acids in most viral isolates) is a transmembrane protein consisting of an N-terminal transmembrane α-helix, followed by a cytoplasmic domain that is likely to be flexible (*Cohen et al., 1988*; *Strebel et al., 1988*). Vpu associates with BST2 through an anti-parallel interaction between the transmembrane domains of each protein (*Mangeat et al., 2009*; *Skasko et al., 2012*; *McNatt et al., 2009*; *Vigan and Neil, 2011*; *Kobayashi et al., 2011*; *Vigan and Neil, 2010*). This interaction is species-specific and essential for the antagonism of BST2 by Vpu (*McNatt et al., 2009*; *Skasko et al., 2012*). Part of Vpu's activity against BST2 can be explained by the viral

**eLife digest** HIV is a retrovirus that attacks the immune system, making the body increasingly susceptible to opportunistic infections and disease and eventually leading to AIDS. While antiretroviral drugs have allowed people with AIDS to live longer, there is no cure or vaccine for HIV.

Two types of HIV exist, with HIV-1 being much more common and pathogenic than HIV-2. Like other 'complex' retroviruses, the HIV-1 genome contains genes that encode various proteins that allow the virus to disrupt the immune response of the host it is attacking. Viral protein u is a protein encoded by HIV-1 (but not HIV-2) that counteracts an antiviral protein called BST2 in the host. BST2, which is part of the host's innate immune response, prevents newly formed viruses from leaving the surface of infected cells. By counteracting BST2, viral protein u allows the virus to spread in the host more efficiently.

Like many proteins, newly produced BST2 is packaged inside structures called vesicles in a part of the cell called the *trans*-Golgi network, and then sent to its destination. Complexes formed by various proteins make sure that the vesicles take their cargo to their correct destinations within the cell. Two adaptor protein complexes—known as AP1 and AP2—are thought to be involved the transport of BST2. However, it is not known how viral protein u stops BST2 from reaching the cell surface, or how it decreases the amount of BST2 in the cell as a whole. Jia et al. show how viral protein u and BST2 jointly interact with AP1. This interaction leads to the mistrafficking and degradation of BST2 and the counteraction of its antiviral activity.

hijacking of the β-TrCP-associated ubiquitin–proteasome degradation pathway (*Van Damme et al., 2008*; *Douglas et al., 2009*; *Goffinet et al., 2009*; *Iwabu et al., 2009*; *Mangeat et al., 2009*; *Mitchell et al., 2009*). A component of the ESCRT-0 machinery, HRS, has also been suggested to recognize ubiquitinated BST2 and target it for lysosomal degradation (*Janvier et al., 2011*). However, these degradation pathways are only partially responsible for the antagonism of BST2 by Vpu (*Van Damme et al., 2008*; *Douglas et al., 2009*; *Iwabu et al., 2009*; *Mangeat et al., 2009*; *Mitchell et al., 2009*). Efficient BST2 downregulation from the cell surface can occur in the absence of BST2 degradation (*Dube et al., 2010*; *Goffinet et al., 2010*; *Tervo et al., 2011*). Moreover, Vpu induces the mistrafficking of BST2 (*Douglas et al., 2009*; *Dube et al., 2010*; *Hauser et al., 2010*; *Lau et al., 2011*; *Schmidt et al., 2011*), causing the accumulation of BST2 at the *trans*-Golgi network (TGN). Both recycled and newly synthesized BST2 are retained at the TGN, blocking the resupply of BST2 to the plasma membrane and eventually leading to its depletion at the cell surface (*Dube et al., 2011*; *Lau et al., 2011*; *Schmidt et al., 2011*). Moreover, mutations in the juxtamembrane hinge region of Vpu that interfere with the localization of Vpu to the TGN impair the antagonism of BST2 (*Dube et al., 2009*; *Vigan and Neil, 2011*).

Clathrin-dependent trafficking pathways have been suggested to be involved in the Vpu-mediated mistrafficking of BST2 (*Kueck and Neil, 2012*; *Lau et al., 2011*; *Mitchell et al., 2009*; *Ruiz et al., 2008*). Such pathways regulate the trafficking of cellular membrane proteins by selectively packaging these membrane cargos into clathrin-coated vesicles (CCV) (*Bonifacino and Traub, 2003*; *Canagarajah et al., 2013*; *Traub, 2009*). The clathrin adaptor protein (AP) complexes mediate this cargo selection. Two canonical sorting motifs in the cytoplasmic domains of the membrane cargo proteins are recognized by the AP complexes: a tyrosine-based Yxxϕ motif (ϕ represents a large hydrophobic residue; x for any amino acid) and an acidic dileucine motif, [E/D]xxxL[L/I]. Five AP complexes exist in the cell and each is responsible for trafficking by distinct routes. For example, AP1 traffics cargo between the TGN and sorting endosomes, while AP2 selects cargo for transport between the plasma membrane and early endosomes (*Canagarajah et al., 2013*). In the absence of Vpu, the natural trafficking of endogenous BST2 depends on the clathrin-associated pathways, and the involvement of both AP1 and AP2 has been suggested (*Rollason et al., 2007*; *Masuyama et al., 2009*). An unusual double-tyrosine motif, YxY, in the BST2 cytoplasmic domain (BST2$_{CD}$) is critical for this natural trafficking (*Rollason et al., 2007*; *Masuyama et al., 2009*). In Vpu, a putative clathrin-sorting motif, ExxxLV (ELV), located in the membrane-distal half of the protein's cytoplasmic domain (Vpu$_{CD}$) was shown to be important for BST2 antagonism (*Kueck and Neil, 2012*, *McNatt et al., 2013*). Furthermore, Vpu-induced virion release and removal of BST2 from the cell surface are inhibited by a dominant negative mutant of AP180, a protein required for the assembly of the CCV at the lipid membrane (*Kueck and Neil, 2012*; *Lau et al., 2011*). However, the critical question remains as to whether any AP complexes, and if so which, are hijacked by Vpu for the downregulation of BST2.

To understand the mechanisms of Vpu-mediated mistrafficking of BST2, we examined the interaction of the cytoplasmic domains of these proteins with recombinant AP complexes and their subunits. Moreover, we determined the crystal structure of a three-component complex containing AP1, the cytoplasmic domain of BST2 ($BST2_{CD}$) and the cytoplasmic domain of Vpu ($Vpu_{CD}$). The structure shows that Vpu mimics a membrane cargo by occupying the acidic dileucine-binding site of AP1, while BST2 is bound at the tyrosine-binding site of AP1. This, together with biochemical and functional evidence, suggests that HIV-1 Vpu is a virally encoded modulator of clathrin-dependent trafficking pathways and supports the involvement of AP1 in the Vpu-mediated mistrafficking of BST2.

## Results

### BST2 specifically binds μ1 of AP1, but not μ2 of AP2 or μ3 of AP3

To identify the AP complexes involved in the trafficking of BST2, we investigated their ability to bind BST2. $BST2_{CD}$ contains a putative tyrosine motif that is believed to bind to the μ subunits of AP complexes, and intracellular interactions between BST2 and the μ subunits of either AP1 or AP2 have been reported (*Rollason et al., 2007*; *Masuyama et al., 2009*). We tested interactions between $BST2_{CD}$ and the C-terminal domains (CTD) of the μ subunits of three AP complexes (AP1, AP2, and AP3). The purified μ1-CTD of AP1 bound $BST2_{CD}$ as the two co-migrated as a higher molecular weight complex on a size exclusion column (*Figure 1A*). In contrast, no such binding was observed for either the μ2-CTD

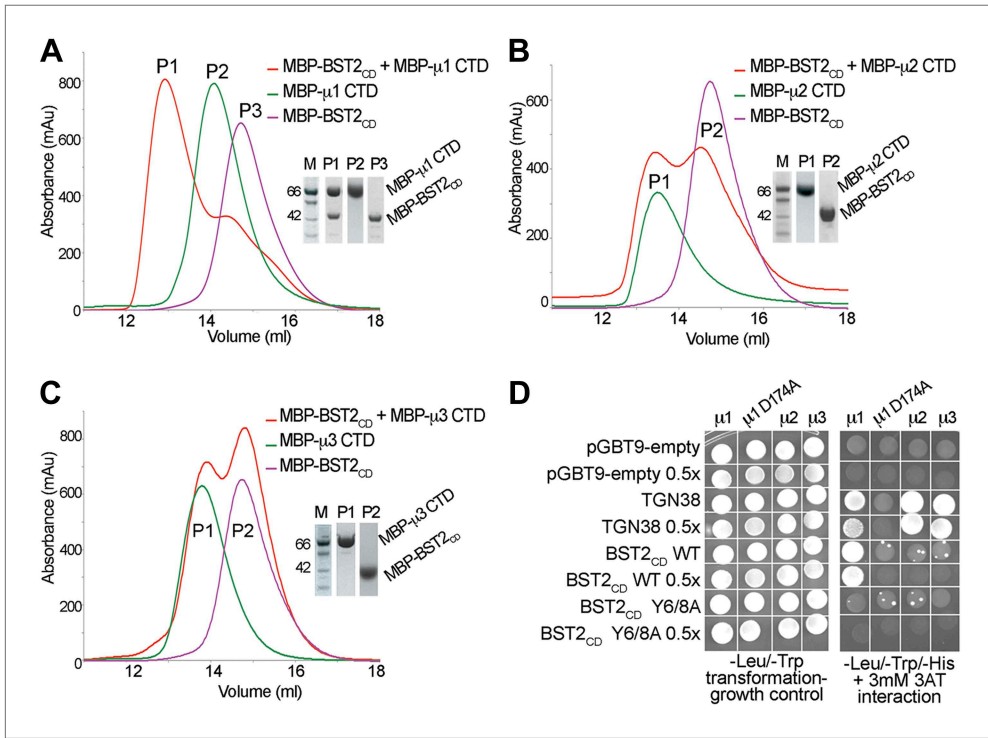

**Figure 1**. BST2 interacts with μ1 but not μ2 or μ3. (**A**–**C**) Size exclusion chromatograms and SDS PAGE analysis of purified MBP-$BST2_{CD}$ (purple curve), MBP-μ CTD (green curve), and their mixture (red curve). A MBP-μ1 CTD and MBP-$BST2_{CD}$ complex is formed in (**A**), as indicated by the shift (P1) of its elution volume from those of the individual components (P2 and P3). No complex formed between BST2 and μ2 (**B**) or μ3 (**C**). (**D**) Yeast 2-hybrid assays showing binding of $BST2_{CD}$ to μ1, but not μ2 or μ3. Growth on -Leu/-Trp/-His + 3 mM 3AT plates indicates interaction. Growth on -Leu/-Trp serves as a growth control. '0.5x' indicates plating of one-half the amount of yeast cells relative to other spots. The $BST2_{CD}$-μ1 interaction is abolished by the tyrosine motif mutation Y6/8A in $BST2_{CD}$ or the tyrosine-binding pocket mutation D174A in μ1.

The following figure supplements are available for figure 1:

**Figure supplement 1**. BST2 does not interact with the α appendage domain of AP2.

(*Figure 1B*) or μ3-CTD (*Figure 1C*). These observations were further confirmed by a yeast two-hybrid (Y2H) assay (*Figure 1D*). Consistent with the in vitro binding assay, BST2$_{CD}$ exhibited binding to μ1, but not μ2 or μ3, in the Y2H assay. As controls, the TGN38 peptide containing a canonical YxxΦ motif displayed binding to all three μ subunits, and a mutant μ1 subunit (μ1 D174A) with a disrupted binding pocket for tyrosine-based motifs bound neither TGN38 nor the BST2$_{CD}$. Furthermore, this binding was abolished by the double alanine mutation of the YxY motif (Y6/8A) in BST2$_{CD}$. These results indicate that BST2$_{CD}$ not only contains a tyrosine-based motif to allow for AP binding, but also a specificity determinant to select specifically for μ1.

Our results support the involvement of AP1 in trafficking of BST2 and are consistent with a previous observation that BST2 binds to the μ1 subunit of AP1, but not μ2 of AP2 (*Masuyama et al., 2009*). In the absence of Vpu, BST2 has been suggested to localize to the TGN through clathrin-dependent trafficking (*Kupzig et al., 2003*; *Rollason et al., 2007*; *Masuyama et al., 2009*), and the BST2–AP1 interaction may be responsible for this localization. In contrast, an involvement of AP2 in BST2 trafficking has been suggested to occur through its α appendage domain. However, this interaction was not detected in our in vitro binding test using size exclusion chromatography (*Figure 1—figure supplement 1*). A specific site on AP2 for BST2 binding remains to be clearly elucidated.

## HIV-1 Vpu binds to multiple subunits of both AP1 and AP2, but not μ3 of AP3

Given the native affinity of BST2 for AP1 and the retention of BST2 in the TGN by Vpu, we sought to define the potential interactions between Vpu and AP1. As the functionally active ELV motif in Vpu$_{CD}$ is a putative acidic dileucine sorting signal, we hypothesized that Vpu binds AP1 at the acidic dileucine-binding site located at the γ and σ1 subunits. To test this, we created a truncated AP1 core with the μ1-CTD removed. Such a truncation construct (AP1t) mimics the open conformation of AP1, exposing the dileucine binding site on AP1 for interaction with cargo (*Jackson et al., 2010*). We co-expressed Vpu$_{CD}$ with casein kinase II (CKII) to phosphorylate Vpu$_{CD}$ at S52 and S56 to mimic its cellular state (*Figure 2—figure supplement 1*). Unless mentioned otherwise, doubly phosphorylated Vpu constructs were used in all in vitro studies. Indeed, MBP-Vpu$_{CD}$ co-migrated with the μ1-truncated AP1 as a complex on a size exclusion column (*Figure 2A*). Importantly, the binding was abolished by the alanine-mutation of the ELV motif, signifying the crucial role of the ELV motif in the interaction between Vpu and AP1 (*Figure 2B*). We further tested the binding between Vpu$_{CD}$ and μ1-CTD and observed complex formation using size exclusion chromatography (*Figure 2C*). These interactions demonstrate that Vpu has evolved the ability to associate with multiple subunits of AP1, potentially allowing it to modulate the cellular trafficking machinery to target host proteins such as BST2.

We next investigated the interaction between Vpu and subunits of AP2 and AP3. We used size exclusion chromatography to test the binding between Vpu$_{CD}$ and a truncated hemicomplex of AP2, α (1–398)-σ, which contains the binding pocket for the acidic dileucine sorting motif; an interaction complex was observed (*Figure 2D*). Furthermore, binding between Vpu$_{CD}$ and the μ2-CTD of AP2 was also observed (*Figure 2E*), while no such binding was observed between Vpu$_{CD}$ and the μ3-CTD of AP3 (*Figure 2F*). Altogether, these results suggest specific interactions between Vpu and AP1 and AP2, which may allow the viral protein to hijack the associated trafficking pathways. However, because BST2$_{CD}$ specifically binds only to μ1, but not to μ2 or μ3 (*Figure 1*), AP1 may play a more significant role than AP2 in the Vpu-mediated antagonism of BST2. This notion is supported by multiple observations, with one exception (*Iwabu et al., 2010*), that Vpu does not increase the rate of BST2 internalization from the cell surface (*Mitchell et al., 2009*; *Dube et al., 2010*; *Andrew et al., 2011*; *Schmidt et al., 2011*).

## Fusion of BST2$_{CD}$ and Vpu$_{CD}$ enhances binding to AP1

The ability of Vpu to interact simultaneously with BST2 and AP1 suggests that Vpu may enhance a native but weak affinity between BST2 and AP1 to increase their binding and consequently retain BST2 in endosomes including the TGN and/or target it to lysosomes. We used a fusion of BST2$_{CD}$ and Vpu$_{CD}$ to mimic in vitro the membrane-assisted binding that occurs in cells. BST2 and Vpu each have a transmembrane (TM) helix through which the two proteins associate. The C-terminus of BST2$_{CD}$ and the N-terminus of Vpu$_{CD}$ are placed close to each other by the interacting TM helices (*Skasko et al., 2012*). This facilitates the convenient design of a 10-amino acid fusion linker that mimics the restraints exerted by the TM helices and links the cytoplasmic domains in an appropriate spatial arrangement (*Figure 3A*).

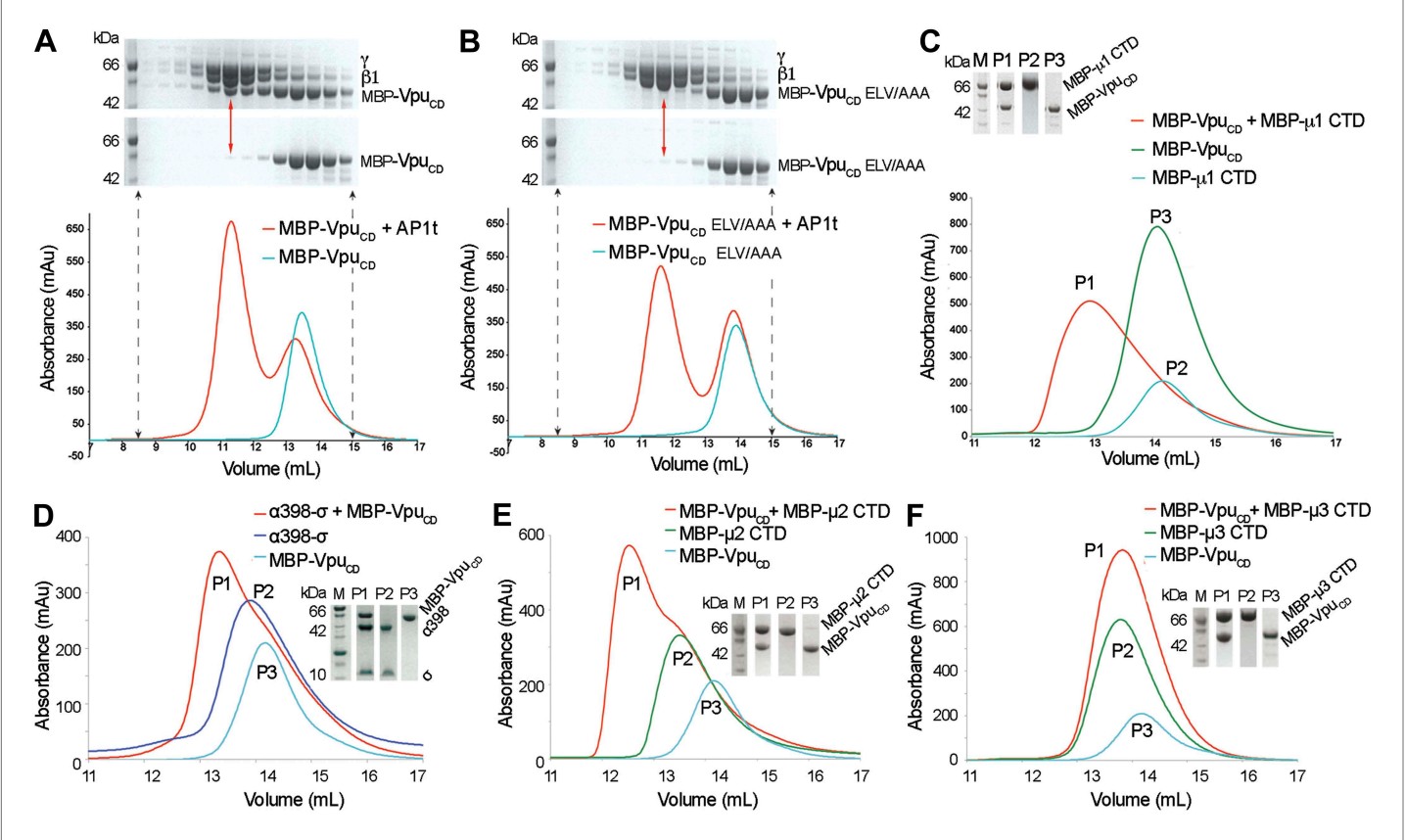

**Figure 2**. Vpu interacts with multiple subunits of AP1 and AP2, but not μ3 of AP3. (**A** and **B**) Size exclusion chromatograms and SDS PAGE analysis of purified MBP-Vpu$_{CD}$ constructs (cyan curve and bottom SDS gel) and its mixture with μ1-truncated AP1 (AP1t) (red curve and top SDS gel). (**A**) The elution profile of MBP-Vpu$_{CD}$ on the size exclusion column is altered in the presence of AP1t (marked by the red arrow), indicating complex formation. (**B**) The alanine mutation of the ELV motif in Vpu$_{CD}$ (MBP-Vpu$_{CD}$ ELV/AAA) abolishes the interaction with AP1t, as the elution profile of MBP-Vpu$_{CD}$ ELV/AAA does not change with or without AP1t. (**C**) MBP-Vpu$_{CD}$ binds to MBP-μ1 CTD. A shift occurs in the elution volume of the MBP-μ1 CTD and MBP-Vpu$_{CD}$ mixture (P1) relative to the individual components (P2 and P3). (**D**) Size exclusion chromatogram and SDS PAGE analysis showing the co-migration of MBP-Vpu$_{CD}$ and α398-σ to a higher molecular weight region relative to the individual components. (**E**) MBP-Vpu$_{CD}$ and MBP-μ2-CTD also form a co-migrating interaction complex. (**F**) No complex is formed between MBP-Vpu$_{CD}$ and MBP-μ3-CTD as the mixture migrates in the same manner as the individual components.

The following figure supplements are available for figure 2:

**Figure supplement 1**. Efficient and complete double phosphorylation of recombinant VpuCD.

---

The fusion protein exhibited strong binding to the GST-tagged AP1 in the pulldown assay (**Figure 3B**). In contrast, although both Vpu$_{CD}$ and BST2$_{CD}$ interact with AP1 subunits in our size exclusion chromatography assays (**Figure 1**, **Figure 2**), their individual interactions with the full AP1 core complex were not observed under stringent pulldown conditions (**Figure 3B**). This result suggests that an additive or cooperative effect occurs when BST2$_{CD}$ and Vpu$_{CD}$ bind to AP1.

## BST2/Vpu/AP1 interaction involves BST2 YxY and the dileucine motif-binding site on AP1, and is independent of Vpu phosphorylation

We tested the requirement of the important motifs in BST2 and Vpu for the binding of AP1. We first used a GST-tagged AP1 core to pull down mutants of the BST2$_{CD}$-Vpu$_{CD}$ fusion (**Figure 4**). As expected, alanine mutation of the YxY motif (Y6/8A) greatly reduced the binding to AP1. Additional mutation of the γ subunit R15E on AP1 that disrupts the acidic dileucine-motif binding site affected the binding further. To validate the in vitro observations made with the TM-free BST2$_{CD}$-Vpu$_{CD}$ fusion, we designed a TM-containing chimera and performed immunoprecipitation experiments using human cells (**Figure 4B,C**). This chimera contains the N-terminal 66 residues of BST2 (including the cytoplasmic domain, TM, and

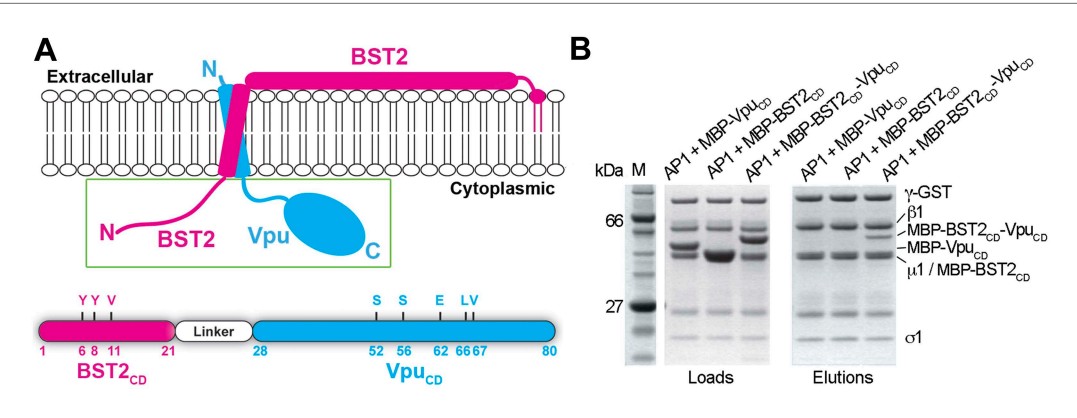

**Figure 3**. BST2$_{CD}$-Vpu$_{CD}$ fusion binds tightly to AP1. (**A**) Schematic of the BST2–Vpu interaction at the lipid membrane (top) and the fusion protein used for in vitro studies (bottom) showing the location of important residues in both BST2 (YxY) and Vpu (ELV). S52 and S56 are phosphorylation sites required for the binding of Vpu to β-TrCP. The regions of Vpu and BST2 used in the fusion are boxed in green. (**B**) GST pulldown assay using an AP1 with a GST-tagged γ subunit. GST bead-bound AP-1 captured MBP-BST2$_{CD}$-Vpu$_{CD}$, but not MBP-Vpu$_{CD}$ or MBP-BST2$_{CD}$ alone. SDS PAGE analysis shows the amount of protein in the load and eluted from the GST beads.

two of the three cysteines in the ectodomain involved in dimerization) followed by a flexible linker (GGGSx3), a FLAG epitope tag, and the entirety of Vpu; we expected it to faithfully recapitulate the geometry of the two interacting molecules in cells (**Figure 4B**). Consistent with the in vitro observations (**Figure 3B**), binding between endogenous AP1 (the μ1subunit) and this chimera was detected, while no such binding was detected for either BST2 or Vpu alone (**Figure 4C**). Although the alanine mutations of either Y6/8 or ELV did not substantially reduce the binding, the mutant combining both sets of mutations was unable to bind AP1 (**Figure 4C**).

Our experiments show that the binding of Vpu to AP1 is independent of serine-phosphorylation (**Figure 4A**). Phosphorylation of S52 and S56 in Vpu is critical for recruiting β-TrCP and its associated E3 ubiquitin ligase complex. The unphosphorylated fusion protein bound AP1 as tightly as the phosphorylated construct (**Figure 4A**). Furthermore, the binding of AP1 was not affected by the double mutation, S52/56N, that destroys the phosphorylation sites (**Figure 4A**). These results are consistent with the notion that the β-TrCP-dependent BST2-degradation by Vpu and the mistrafficking of BST2 by Vpu are governed by different determinants in the Vpu$_{CD}$.

## Crystal structure of BST2 and Vpu binding to the open AP1 core

To understand the Vpu-enhanced binding of BST2 to AP1, we determined the crystal structure of the BST2$_{CD}$-Vpu$_{CD}$ fusion in complex with the AP1 core at 3.0 Å resolution (**Figure 5A**). The AP1 core adopts an activated, open conformation, with both of its cargo binding sites exposed for interaction with BST2$_{CD}$-Vpu$_{CD}$. The AP1 in the current structure adopts an open conformation distinct from that observed previously for AP2 and AP1 (**Jackson et al., 2010**; **Ren et al., 2013**). BST2$_{CD}$ binds to the tyrosine motif-binding site on AP1 through critical interactions involving the YxY motif, while Vpu$_{CD}$ occupies the acidic dileucine motif-binding site of AP1 through the ELV motif. Only a short region of Vpu$_{CD}$ flanking the ELV motif is well ordered and successfully built in the structure. There is no direct interaction between BST2$_{CD}$ and Vpu$_{CD}$. Overall, the structure reveals how Vpu enhances the native interaction between BST2 and AP1. By combining the viral protein's affinity for AP1 via the ELV motif and the tight transmembrane interaction with the host protein, Vpu appears to act as an adaptor to increase the affinity of AP1 for BST2.

## BST2 binds to AP1 via an unusual YxY motif

BST2$_{CD}$ occupies the conserved tyrosine motif-binding site on the μ1 subunit of AP1 (**Figure 5B–D**). However, the observed interface differs from the canonical tyrosine peptide (YxxΦ) binding in that the interaction is achieved with an unusual double-tyrosine motif (YxYxxΦ) (**Figure 5B,D**). Y8 of BST2$_{CD}$ forms the canonical interactions with μ1 residues and inserts into the conserved tyrosine-binding

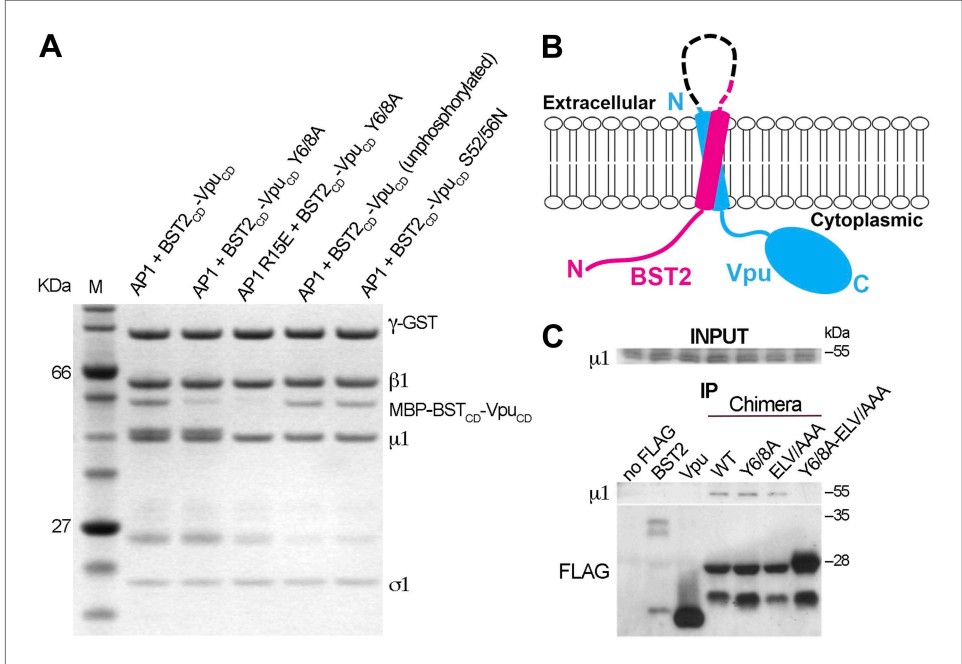

**Figure 4**. BST2/Vpu/AP1 interaction involves the BST2 YxY sequence and the dileucine motif-binding pocket on γ/σ1 and is independent of Vpu phosphorylation. (**A**) GST affinity pulldown assay using an AP1 core complex with a GST-tagged γ subunit. Mutation of the acidic dileucine pocket on γ ('AP1 R15E') and the YxY motif on BST2 substantially reduced binding to AP1. Phosphorylation or S52/56N mutation of Vpu has no effect on AP1 binding. (**B**) Schematic of the TM-containing BST2 and Vpu fusion chimera. The broken line indicates the first 20 residues of the native BST2 ectodomain followed by the fusion linker, the FLAG epitope tag, and the Vpu ectodomain. (**C**) HEK293T cells were transfected to express FLAG-tagged Vpu, BST2, or the fusion chimera or related mutants. 'no FLAG' indicates mock transfection without any FLAG-tagged construct. Notably, Vpu is much more abundantly expressed than either BST2 or the BST2-Vpu chimeras, and the chimeras appear to be partly proteolyzed. Nonetheless, immunoprecipitation (IP) using anti-FLAG antibody shows endogenous μ1 only in the presence of the BST2-Vpu chimera. Mutation of both the BST2 YxY and Vpu ELV motifs is necessary to prevent co-IP of μ1.

pocket on μ1. Unlike the canonical cases in which a leucine or isoleucine residue is found at the Y+3 position, V11 in BST2 only partly fills the corresponding hydrophobic pocket on μ1 (**Figure 5C**). This pocket would be better satisfied by a larger hydrophobic residue. This observation is consistent with results from a previous combinatorial screen of tyrosine-based μ-binding sequences: for μ1, valine was disfavored at the Y+3 position, whereas leucine was favored (**Ohno et al., 1998**). The relatively modest interaction mediated by V11 is presumably compensated by Y6, which stabilizes the binding through hydrogen-bonding interactions with N308 and E381 of μ1 and through stacking interactions with μ1 P383 and Y384. Consistent with the notion that Y6, Y8, and V11 each contribute to the overall binding, substitution of any of these single residues with alanine was sufficient to disrupt the interaction between BST2$_{CD}$ and μ1 as detected using the Y2H assay (**Figure 5E**).

The structure also explains why BST2, specifically the YxYxxV sequence, has a preference for μ1, but not μ2 or μ3 (**Figure 1D**). An overlay of the BST2$_{CD}$-bound μ1 structure to the μ2 structure shows that μ2 lacks a tyrosine residue, corresponding to μ1 Y384, which provides a stabilizing stacking interaction with BST2 Y6 (**Figure 5—figure supplement 1A**). In addition, the presence of μ2 Q318, in place of the smaller N308 of μ1, disrupts a hydrogen-bonding interaction with the side chain of BST2 Y6 and potentially causes steric hindrance. In the case of μ3, the structural difference is much more pronounced: severe steric clashes appear to prevent the binding of BST2 Y6 (**Figure 5—figure supplement 1B**). Overall, these results not only explain a critical feature of Vpu-mediated hijacking of the AP1-dependent CCV pathway for BST2 antagonism, but also serve as the first structural example of a cellular YxY-based sorting signal bound to the μ1 subunit of AP1.

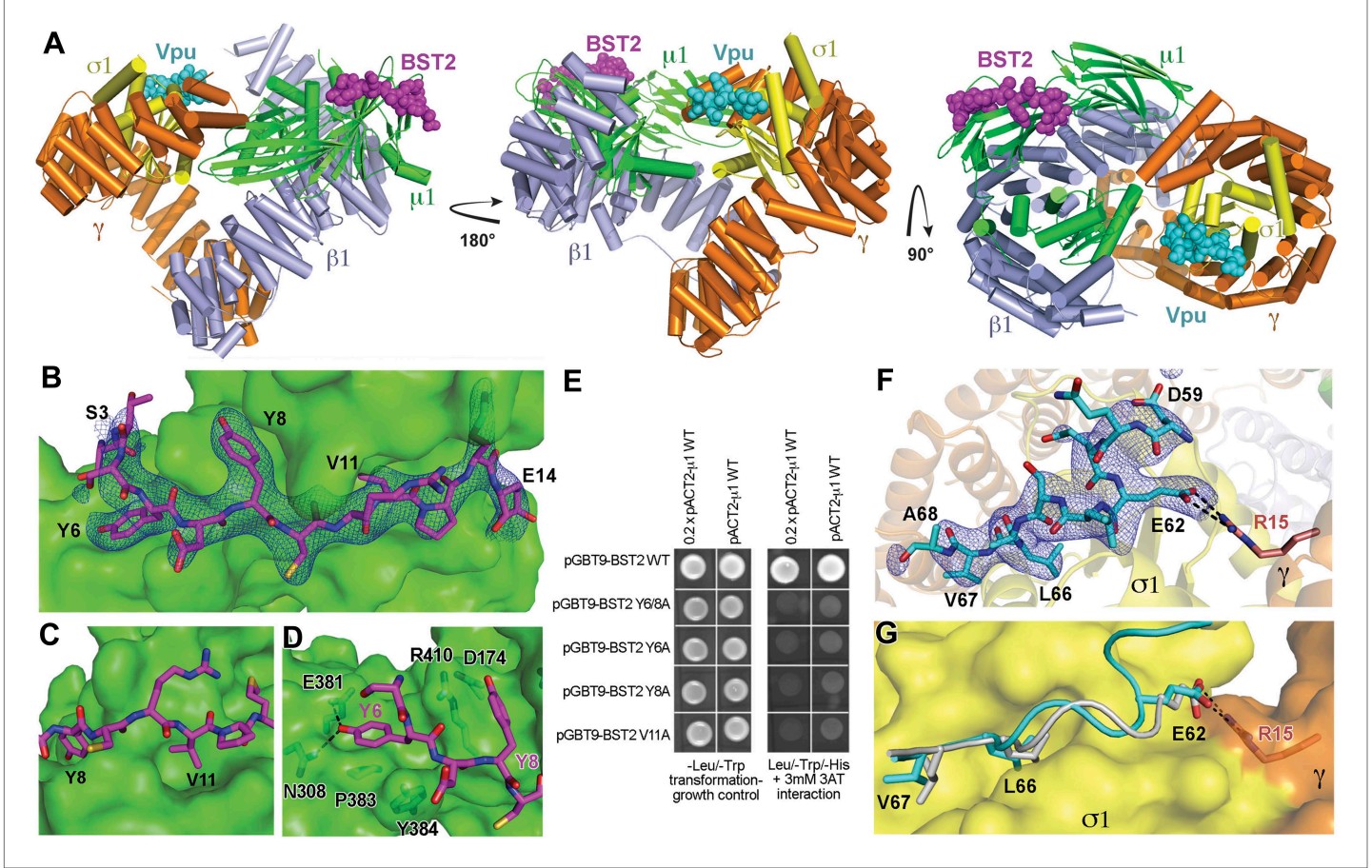

**Figure 5**. Crystal structure of the BST2$_{CD}$-Vpu$_{CD}$/AP1 complex. (**A**) The crystal structure of the BST2$_{CD}$-Vpu$_{CD}$/AP1 complex. AP1 is colored by subunit (β1 in gray, γ in orange, μ1 in green and σ1 in yellow). Vpu$_{CD}$ (cyan) binds to the acidic dileucine binding pocket of γ/σ1, and BST2$_{CD}$ (magenta) binds to the tyrosine-binding pocket in μ1. (**B**) The difference Fourier map (mFo-DFc at 3σ level, blue mesh) of BST2$_{CD}$ (magenta sticks) binding to μ1 (green surface). Important residues in BST2$_{CD}$ are labeled. (**C**) V11 partly fills the canonical Φ residue binding-site on μ1. (**D**) Y6 and Y8 make extensive interactions to μ1 residues. Hydrogen bonds are indicated with dashed lines. (**E**) Yeast 2-hybrid assays showing binding of BST2 to μ1. Growth on -Leu/-Trp/-His + 3 mM 3AT plates indicates interaction. Growth on -Leu/-Trp serves as a growth control. '0.2x' indicates plating of one-fifth the amount of yeast cells relative to other spots. The BST2-μ1 interaction is abolished by the alanine mutations of BST2 Y6/8A, Y6A, Y8A, or V11A. (**F**) The difference Fourier map (mFo-DFc at 3σ level, blue mesh) of Vpu$_{CD}$ (cyan sticks) binding to σ1 (yellow surface) and γ (orange surface) subunits of AP1. Important Vpu$_{CD}$ and γ residues are labeled. (**G**) Overlay of α/σ2 of AP-2 (PDB ID 2JKT) (*Kelly et al., 2008*) and γ/σ1 of AP1 shows that the Vpu ELV motif (cyan) binds in the same way as the canonical peptide (gray).

The following figure supplements are available for figure 5:

**Figure supplement 1**. Structural incompatibility prevents the binding of BST2 to either μ2 of AP2 or μ3 of AP3.

## Vpu binds to AP1 by mimicking a canonical acidic dileucine motif

Opposite to the BST2-binding site on the activated AP1 core, Vpu$_{CD}$ associates with the γ and σ1 subunits of AP1 by mimicking the canonical acidic dileucine-sorting motif (*Figure 5*). E62 forms a salt bridge with R15 of AP1 γ subunit, fulfilling the role of the 'acidic residue' within the sorting motif, while L66 and V67 embed into the hydrophobic pocket on AP1 σ1 that accommodates the canonical dileucine residues (*Figure 5F*). The Vpu ELV motif overlays closely with a canonical acidic dileucine-sorting motif when bound to AP2 (*Figure 5G*). Our results reveal that the ELV motif of Vpu acts as a sorting motif mimic for hijacking AP1 and thus the CCV pathway for mistrafficking of BST2.

## A novel open conformation is observed for the BST2/Vpu-activated AP1

While the previously observed Arf1-bound AP1 exhibits the same level of opening as the activated AP2 (*Jackson et al., 2010*; *Ren et al., 2013*), the AP1 in the BST2/Vpu-bound structure adopts a

conformation that is much more open than the previously observed structures. When the BST2/Vpu-activated AP1 structure was overlaid with the Arf1-activated AP1 structure using the β1 subunits, a twisting of the γ and σ1 subunits was observed (*Figure 6A*), which further exposes the dileucine-binding pocket at the γ/σ1 interface in AP1. The conformational change involves a ~20° rotation of γ/σ1 around an axis at the base of γ where it contacts β1, with the largest Cα movement of ~35 Å at the tip of γ (*Figure 6A*). Of note, however, the relative positions either between β1 and μ1 or between γ and σ1 are well maintained. As a result of the γ/σ1 movement, new interactions occur between the μ1-CTD and the N-terminal portion of γ. This new γ–μ1 interface has a buried surface area of 685 Å², and consists of extensive, hydrogen-bonding and salt bridge interactions between the two subunits (*Figure 6B,C*). We further created a double mutation, γ Q28R and μ1 D319R, to disrupt this new interface and tested its role in the binding of AP1 to BST2/Vpu in the GST pulldown assay. The binding was decreased, indicating that this newly observed γ–μ1 interface is important for the Vpu-mediated manipulation of AP1 (*Figure 6D*). This new AP1 conformation supports the recent hypothesis that the AP complexes might be able to access a wider conformational space beyond what has been previously observed in the locked, unlatched, and open states (*Canagarajah et al., 2013*).

## Vpu R44 and L/I45 potentially interact with the μ1 subunit of AP1

The newly formed γ–μ1 interface leads to local structural changes. Specifically, the μ1 loop from P363 to G372, which is disordered in the previous AP1 structures (*Heldwein et al., 2004*; *Jia et al., 2012*; *Ren et al., 2013*), became well ordered in the current structure. Although all of the nearby residues from both γ and μ1 are well ordered, considerable additional electron density remains near the γ–μ1 interface, suggesting the possible presence of Vpu residues (*Figure 7A*). Although the quality of the additional electron density does not permit model building, we suspect it may belong to a portion of VpuCD that is N-terminal to the ELV, judging by the location of the electron density relative to Vpu ELV (*Figure 7A*). Specifically, Vpu residues R44 and L45 (NL4-3 sequence, corresponding to I45 of Vpu

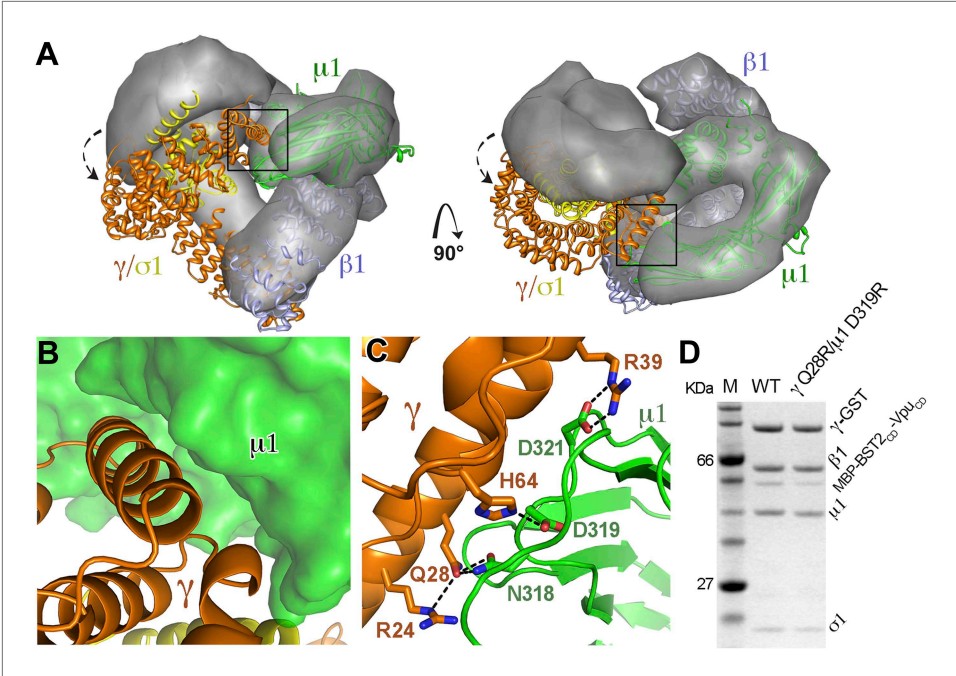

**Figure 6**. AP1 adopts a novel open conformation when bound to BST2CD and VpuCD. (**A**) Overlay of the Arf1-bound AP1 structure (PDB ID 4HMY, grey surface) (*Ren et al., 2013*) and the BST2CD-VpuCD-bound AP1 structure (ribbon representation with β1 in blue, γ in orange, μ1 in green and σ1 in yellow). The broken arrows point to the large relative movement of the γ/σ1 subunits of our structure with respect to Arf-1/AP1. Note that the β1/μ1 subunits overlay well in the two structures. (**B** and **C**) Close-up views of the new γ-μ1 interface observed in the present study, boxed in (**A**), with hydrogen bonds represented by dashes. (**D**) GST affinity pulldown assay using an AP1 core complex with GST-tagged γ subunit. Mutations at the γ-μ1 interface reduced MBP-BST2CD-VpuCD binding to AP1.

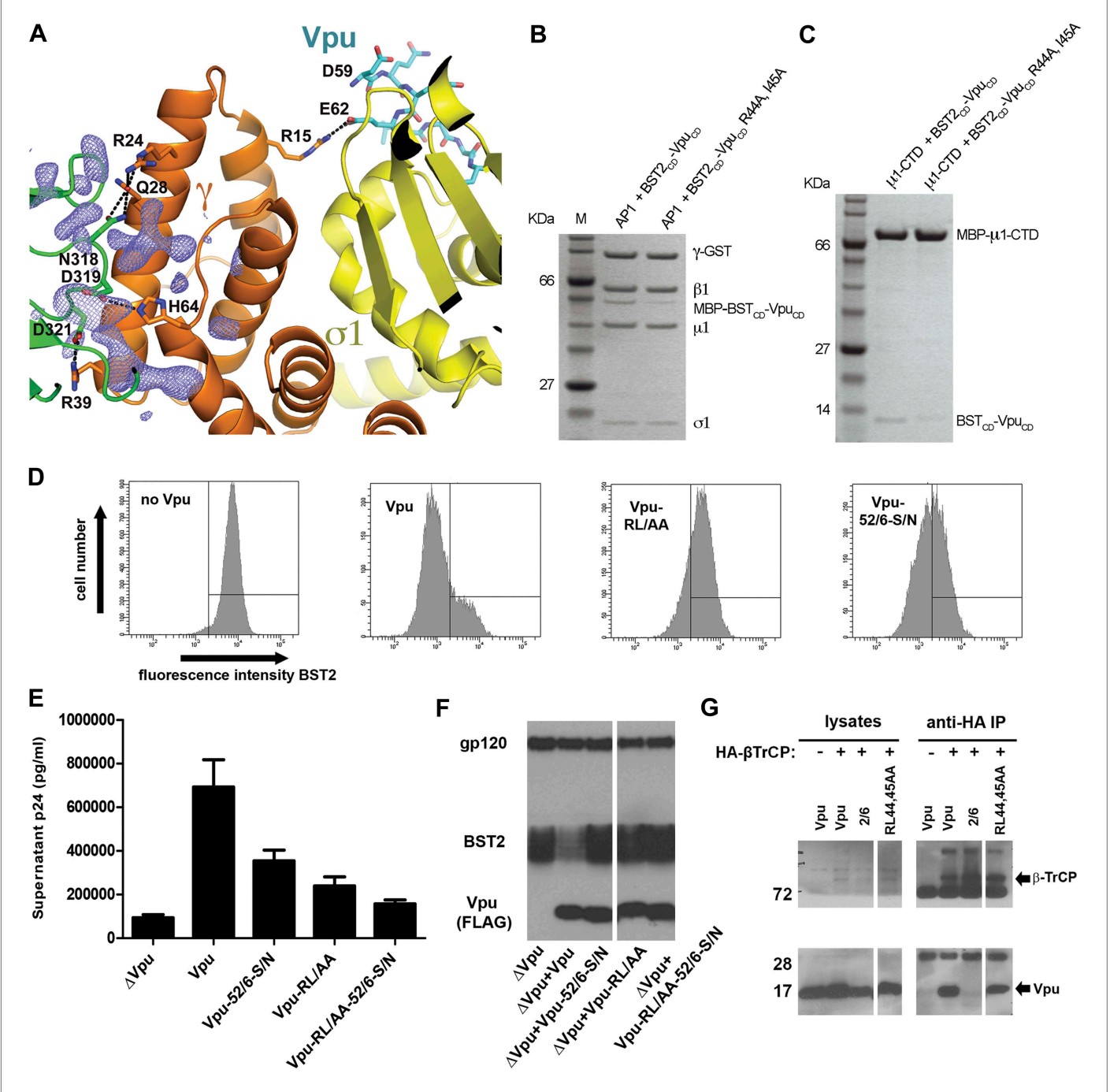

**Figure 7**. Vpu R44, L/I45 residues may interact with AP1 at the γ-μ1 interface and contribute to the optimal Vpu activity. (**A**) The difference Fourier map (mFo-DFc at 3σ level, blue mesh) near γ (orange)-μ1 (green) interface of AP1. VpuCD is shown in cyan sticks and AP1 σ1 is in yellow. (**B** and **C**) Affinity pulldown assays using an AP1 with GST-tagged γ subunit (**B**) or using a His-tagged MBP-μ1-CTD (**C**). SDS PAGE analysis shows the amount of protein in the load and elution. Mutation of these potentially important Vpu residues at the γ-μ1 interface, R44A:I45A, significantly affected the binding interactions. (**D**) Downregulation of surface BST2 by Vpu is impaired by the R44A:L45A (RL/AA) mutation as well as by the S52N:S56N (S52/56N) mutation. (**E**) HEK293T cells were co-transfected to express provirus lacking *vpu*, and the indicated codon-optimized Vpu proteins. The Vpu R44A:L45A and S52N:S56N mutations each impair Vpu mediated-virion release, and their effects are additive. (**F**) Western blot of the experiment shown in **E**, indicating the expression levels of BST2 and Vpu; gp120 is the viral envelope glycoprotein. (**G**) Co-IP showing that β-TrCP binding was abolished by the S52/56N ('2/6') mutation, but not the RL/AA mutation.

The following figure supplements are available for figure 7:

**Figure supplement 1**. VpuCD exhibits large conformational flexibility.

HV1S1 used in the structural study) upstream of ELV may participate in the interaction with AP1, as these conserved residues have been implicated in Vpu-mediated antagonism of BST2 (*Lucas and Janaka, 2012*; *Pickering et al., 2014*).

We tested the Vpu R44A and I45A double mutation for its effect on the binding to AP1 in vitro. The binding was impaired by this mutation (*Figure 7B*). Specifically, the mutation substantially reduced the binding between the BST2$_{CD}$-Vpu$_{CD}$ fusion and μ1-CTD of AP1 in our pulldown assay (*Figure 7C*). This interaction likely explains the observed affinity between Vpu and μ1 (*Figure 2C*). It also suggests that the extra electron density in our structure (*Figure 7A*) may come from Vpu residues including R44 and L/I45. These Vpu residues may interact with γ/μ1 subunits of AP1 and stabilize γ–μ1 contacts and the novel AP1 conformation.

We characterized the effect of the R44A:L45A mutation on the activity of Vpu as an antagonist of BST2 in human cells. The R44A:L45A mutation significantly impaired the ability of Vpu to reduce the amount of BST2 at the cell surface (*Figure 7D*). It also greatly impaired the ability of Vpu to enhance virion release (*Figure 7E*). Interestingly, the effect of the R44A:L45A mutation on virion release was additive with the S52/56N mutation, which ablates the binding of Vpu to β-TrCP (*Figure 7E,G*), despite that both mutations impair the apparent degradation of BST2 (*Figure 7F*). These data are consistent with the proposed role of R44 and I45 in the binding of Vpu to AP1 rather than to β-TrCP. Indeed, the R44A:L45A mutation did not affect the interaction with β-TrCP as measured by immunoprecipitation (*Figure 7G*). Together, the impaired abilities of Vpu R44A:L45A to bind AP1 and to decrease the steady-state expression of BST2 suggest that AP1 is important for the Vpu-mediated endo-lysosomal degradation of BST2.

We note that to achieve the interaction of R44, L/I45 with μ1 and the interaction of the ELV sequence with γ-σ1, Vpu$_{CD}$ must adopt an extended conformation. Although observed with considerable helical content in previous NMR studies, Vpu$_{CD}$ was believed to be flexible (*Figure 7—figure supplement 1*) (*Willbold et al., 1997*; *Wittlich et al., 2009*). Furthermore, these secondary structures were observed under conditions that induce helix formation. Specifically, the helical feature of the first half of Vpu$_{CD}$ was shown to be relatively more pronounced, whereas the latter helix, harboring the ELV sequence, exhibited low helical content and was more likely to be unstructured (*Wittlich et al., 2009*). Our study provides further evidence for the flexible nature of Vpu$_{CD}$.

## BST2 YxY and Vpu ELV motifs are both required for the optimal enhancement of virion release by Vpu and for the Vpu-mediated decrease in the expression of BST2

We performed virion release assays to confirm the functional requirements of the BST2 YxY and Vpu ELV motifs in human cells (*Figure 8*). The mutation Y6/8A in BST2$_{CD}$ markedly impaired the ability of Vpu to promote virion release, supporting a critical role for this trafficking motif in antagonism of BST2 by Vpu. Since the BST2 YxY motif is not likely required for the β-TrCP-dependent degradation pathway,

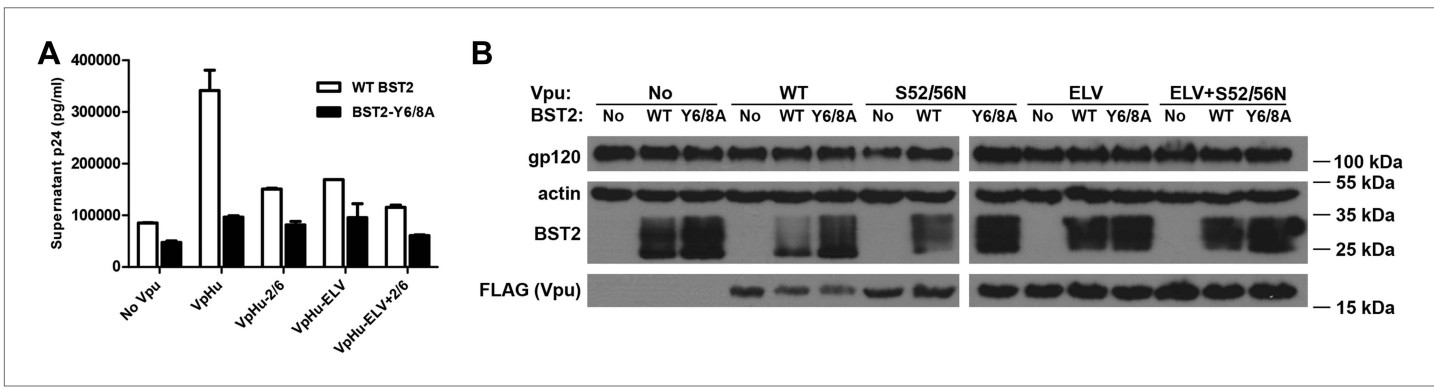

**Figure 8**. The BST2 YxY as well as the Vpu ELV and di-serine motifs are each important for Vpu-mediated reduction in BST2-expression and for optimal virion release. (**A**) HEK293T cells were co-transfected to express the indicated BST2 variants, provirus lacking *vpu*, and the indicated codon-optimized Vpu proteins. Mutations of the BST2 YxY, Vpu ELV and S52/S56 ('2/6') sequences have additive effects in decreasing the efficiency of Vpu mediated-virion release. (**B**) Western blot showing the expression levels of BST2 (WT or Y6/8A) and Vpu (WT or mutants).

which functions via ubiquitination, its importance likely comes from its affinity to AP1 and clathrin-dependent pathways. Of note, the Y6/8A mutant was better expressed than wild-type BST2 and restricted virion release more effectively both in the presence and absence of Vpu. Alanine mutation of Vpu ELV also impaired virion release, to a degree similar to that of the S52/56N mutation of the β-TrCP binding site. Notably, mutation of the ELV sequence in Vpu and the YxY sequence in BST2 increased the steady-state expression of BST2, presumably by inhibiting endo-lysosomal degradation either as it occurs natively or as stimulated by Vpu. This scenario is consistent with our structural and biochemical observations that both BST2 YxY and Vpu ELV motifs interact with AP1, and it further supports the hypothesis that this interaction is part of the endo-lysosomal degradation mechanism that supports Vpu-activity.

## BST2 YxY and Vpu ELV motifs contribute to the co-localization of BST2 and Vpu with AP1 at the cell center

We used immunofluorescence microscopy to verify the co-localization of BST2, Vpu and AP1 at the juxtanuclear region near the cell center (*Figure 9*; *Figure 9—figure supplement 1*). In HeLa cells that express BST2 constitutively, BST2 and AP1 co-localized even in the absence of Vpu, and all three proteins co-localized when Vpu was expressed (*Figure 9—figure supplement 1*). As expected, this co-localization was not affected by the Vpu S52/56N mutation, consistent with our in vitro observation that the β-TrCP binding motif of Vpu is not required for interaction with AP1. We further performed the immunofluorescence experiments after stably transfecting HT1080 cells to express BST2 (allowing the analysis of mutants since these cells do not naturally express BST2) followed by transient transfection to express Vpu. As was seen using the HeLa cells, BST2 and AP1 co-localized in the absence of Vpu, and all three proteins co-localized at the cell center when Vpu was expressed (*Figure 9A*). Mutation of the YxY sequence of BST2, and to a lesser extent the ELV sequence of Vpu, caused mislocalization of the proteins from the cell center region into more peripheral puncta, although some overlap at the cell center with AP1 persisted. Quantitative image analyses indicated that the YxY sequence of BST2 contributes to that protein's co-localization with AP1, as does the ELV sequence of Vpu (*Figure 9B*).

To clarify the role of these sequences in the trafficking of BST2–Vpu complexes, we analyzed the subcellular localization of our BST2–Vpu chimera that includes each protein's transmembrane domain (*Figure 4B*). This chimera has the advantage of 'forcing' the interaction between the two proteins, allowing the experiment to follow the fate of the chimera as a surrogate for BST2/Vpu complexes, without a potentially large background of the individual, uncomplexed proteins. The wild-type chimera localized to the cell center with AP1 as expected (*Figure 9C*). In contrast, the Y6/8A-ELV/AAA mutant colocalized less well with AP1; moreover, it was dispersed into peripheral endosomes and highlighted the plasma membrane. Correspondingly, enhanced surface expression was detected for the mutant chimera by flow cytometry (*Figure 9D*), indicating the loss of Vpu's ability to downregulate BST2 from the cell surface. Overall, these data support the proposed roles of the BST2 YxY and Vpu ELV sequences in the hijacking of AP1 by Vpu for the mistrafficking of BST2.

## Discussion

Mistrafficking of host immune molecules via clathrin-dependent pathways is a strategy frequently employed by primate lentiviruses (*Tokarev and Guatelli, 2011*). For example, hijacking of AP-mediated trafficking pathways is well documented for the viral protein Nef (*Piguet et al., 1999*; *Roeth and Collins, 2006*; *Arhel and Kirchhoff, 2009*). Nef binds AP1 to prevent MHC-I from reaching the cell surface, providing evasion of immune surveillance by cytotoxic T cells (*Collins et al., 1998*; *Le Gall et al., 1998*; *Roeth et al., 2004*; *Noviello et al., 2008*). Nef also engages AP2 to remove the primary viral receptor, CD4, from cellular membranes to avoid super-infection, interference with virion release and infectivity, and the exposure on the cell surface of CD4-induced epitopes within the viral Env glycoprotein (*Garcia and Miller, 1991*; *Foti et al., 1996*; *Craig et al., 1998*; *Chaudhuri et al., 2007*; *Veillette et al., 2013*; *Pham et al., 2014*). In SIV, Nef uses AP2 to remove BST2 from the cell surface to allow for the efficient release of progeny virions (*Zhang et al., 2011*; *Serra-Moreno et al., 2013*). In addition, the HIV-2 Env protein antagonizes BST2 through an AP2-dependent mistrafficking mechanism (*Hauser et al., 2010*).

The work presented herein reveals that HIV-1 Vpu is another viral modulator of host membrane trafficking pathways, specifically, for the AP1-mediated mistrafficking of BST2. This is conceptually

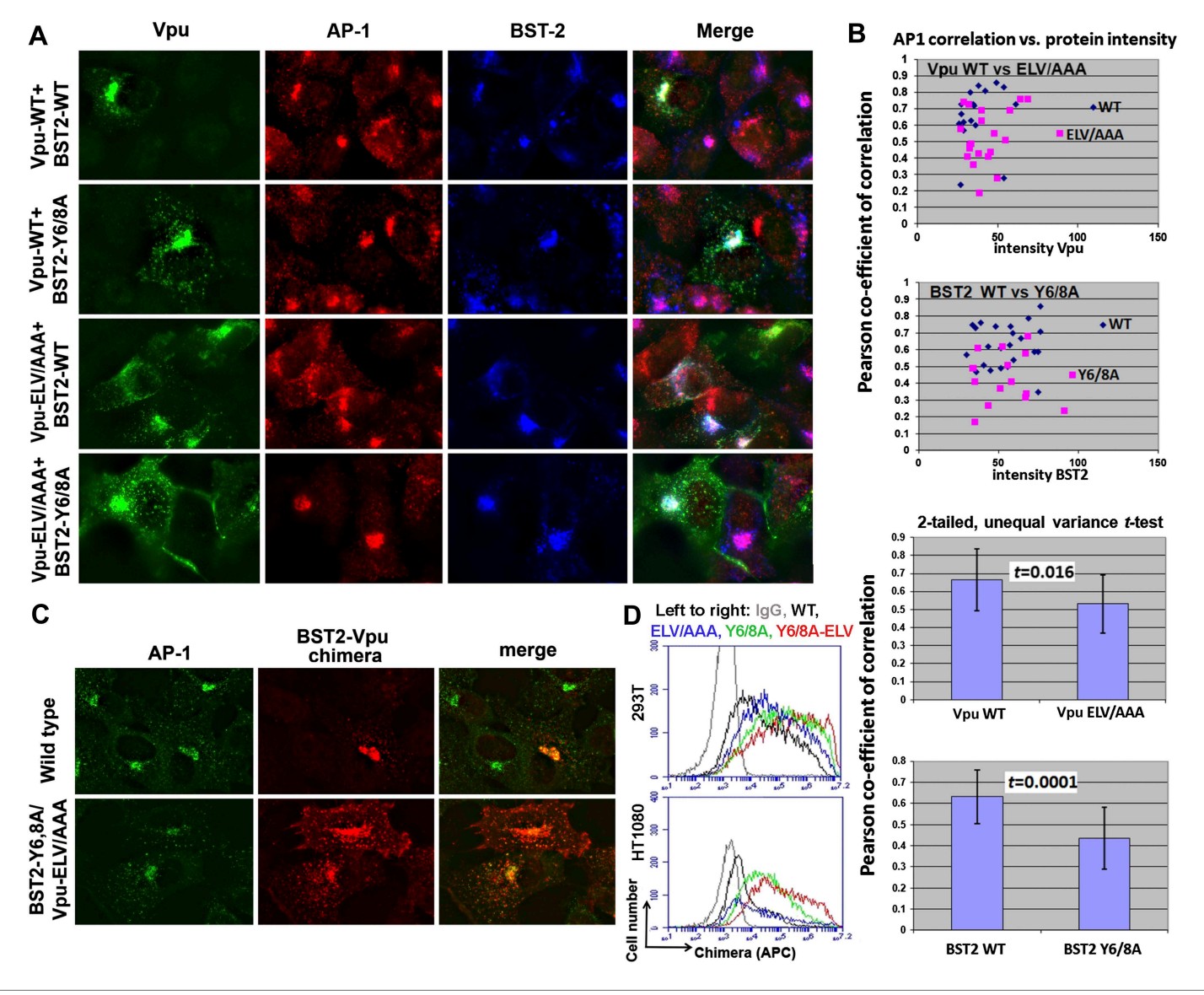

**Figure 9**. YxY and ELV motifs contribute to BST2/Vpu/AP1 co-localization and the trafficking of BST2/Vpu complexes. (**A**) HT1080 cells stably expressing BST2 WT or the Y6/8A mutant were transfected to express Vpu or the ELV/AAA mutant. The cells were stained for Vpu, BST2, and AP1 (γ subunit). (**B**) Pearson co-efficient of correlation between AP1 and Vpu or between AP1 and BST2 vs the intensity of each protein. Each datum point represents a single cell. The correlation between Vpu and AP1 was analyzed in cells expressing wild type BST2, whereas that between BST2 and AP1 was analyzed in cells that were microscopically negative for Vpu expression (although transfected with the wild type Vpu-expression construct). Mutation of the YxY sequence in BST2 and the ELV sequence in Vpu decrease each protein's overlap with AP1. Statistical significance of these data is shown on the bottom. (**C**) HT1080 cell were transfected to express the BST2-Vpu chimera. The wild type chimera co-localized with AP1 at the perinuclear region. Alanine mutations of the YxY and ELV motifs decreased the co-localization of the BST2-Vpu chimera and AP1 and caused displacement of the chimera to peripheral endosomes and the plasma membrane. (**D**) The surface expression of the BST2-Vpu chimera after expression in HEK293T or HT1080 cells by transient transfection was measured using anti-FLAG antibody and flow cytometry. GFP was expressed from a separate plasmid as a transfection marker. The histograms show the cell number vs the intensity of surface stain for the chimera in the GFP-positive (transfected) cells. The Y6/8A mutation caused increased surface expression, as did the ELV/AAA mutation, although to a lesser extent. The highest expression at the cell surface was observed in case of the mutant containing both Y6/8A and ELV/AAA mutations.

The following figure supplements are available for figure 9:

**Figure supplement 1**. Co-localization of BST2, Vpu, and AP1 in HeLa cells transfected to express either the wild type or the S52N/S56N mutant of Vpu.

similar to the HIV-1 Nef-mediated MHC-I downregulation but occurs through different interaction mechanisms with AP1 (*Figure 10*). In each case, the HIV-1 protein, Nef or Vpu, causes the retention of its cellular target, MHC-I or BST2, at the TGN and eventually leads to lysosomal degradation through late endosomal pathways. Nef builds upon an incomplete tyrosine-based sorting motif in the MHC-I cytoplasmic domain (CD) and promotes a cooperative three-protein interaction involving MHC-I CD, Nef and μ1 of AP1 (*Figure 10B*) (*Noviello et al., 2008*; *Wonderlich et al., 2008*; *Singh et al., 2009*; *Jia et al., 2012*). The binding mode for the three-component complex involving BST2, Vpu and AP1, is different, with no three-protein interface. The complex is instead stabilized by pair-wise binary interactions: BST2 and Vpu bind through the transmembrane regions, $BST2_{CD}$ binds the AP1 μ1 subunit, and $Vpu_{CD}$ binds the AP1 σ1/γ/μ1 subunits (*Figure 10A*). This model is consistent with the previous findings that the transmembrane interaction between BST2 and Vpu is of pivotal importance in determining the activity of Vpu against BST2. The BST2/Vpu/AP1 interaction also causes an open conformation of AP1 that has not been observed before. This new AP1 conformation might only be induced by Vpu or alternatively it might exist as a physiologically functional state that Vpu selectively uses to its advantage.

The Vpu-mediated antagonism of BST2 appears to be a multifaceted process. Besides AP1-mediated mistrafficking and degradation, Vpu can also induce the degradation of BST2 through β-TrCP-mediated ubiquitination and the ESCRT machinery. Abolishing either the ubiquitination pathway by the S52/56N mutation in Vpu or the mistrafficking pathway by the ELV/AAA mutation each led to defects in the enhancement of virion release (*Figure 8*) (*Kueck and Neil, 2012*; *McNatt et al., 2013*). Combining the two sets of mutations, however, appeared to cause the most substantial loss of Vpu function, suggesting a potentially parallel nature for these pathways. How these pathways, ubiquitination and AP1-mediated mistrafficking, might work together to enable the antagonism of BST2 by Vpu remains to be further elucidated in finer temporal and spatial details.

The complexity of these trafficking and degradation mechanisms may have prevented a clear identification of the AP complex(es) responsible for the Vpu-activity in previous studies (*Kueck and Neil, 2012*). Despite the evidence supporting the involvement of clathrin-associated pathways, neither AP1, 2 or 3 knockdown in 293T cells and HeLa cells nor AP1 γ knockout in mouse fibroblasts had any apparent effects on the ability of Vpu to antagonize restriction by BST2 (*Kueck and Neil, 2012*), although we previously reported a role for AP2 in the Vpu-mediated surface-downregulation of BST2 (*Mitchell et al., 2009*). Conceivably, the individual roles of a given pathway are obscured by compensation from overlapping parallel pathways, such as the involvement of multiple AP complexes or the monomeric clathrin adaptor HRS. Another possibility is redundancy in the composition of an individual AP complex. A specific subunit of AP could be knocked down or out only to be replaced by a different isoform (*Boehm and Bonifacino, 2001*). Substitution might also happen between two related subunits

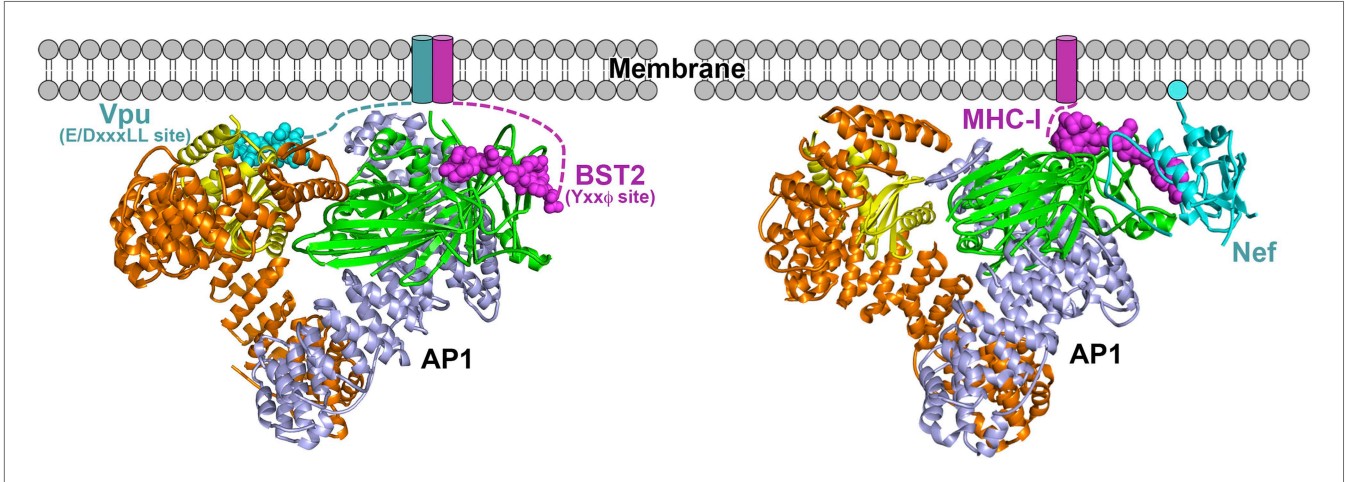

**Figure 10**. Schematics of hijacking of AP1 by HIV-1 Vpu to target BST2 (left) and by HIV-1 Nef to target MHC-I (right) (*Jia et al., 2012*). Transmembrane helices are represented by cylinders and the Nef myristoyl anchor is represented by a cyan sphere.

from different AP complexes as they have highly conserved structures (*Keyel et al., 2008*; *Li et al., 2010*). Another level of complexity might come from the potential for the mechanism of Vpu activity to be at least partly cell-type dependent. For example, BST2 in some cell types appears to be equally vulnerable to β-TrCP-dependent degradation and to mistrafficking, whereas in others the role of β-TrCP-dependent degradation is less important, possibly due to lower levels of BST2 expression (*Schindler et al., 2010*).

The intrinsic, structural flexibility of Vpu may enable its functional versatility and underlie the complexity of BST2-antagonism. A shared feature of retroviral accessory proteins including Vpu, Nef, Vif, Vpr and Vpx is the presence of long and flexible loops that provide the structural basis for their multifunctional nature (*Xue et al., 2012*). These unstructured regions can adopt different conformations to facilitate binding to different host target proteins. These regions often harbor motifs mimicking functional host sequences, enabling the virus to hijack the desired host machineries (*Kadaveru et al., 2008*). The binding of the Vpu ELV motif to AP1 revealed in our structure, as well as that of the di-phosphoserine motif of Vpu to β-TrCP shown before, serve as two excellent examples of this viral strategy.

The ExxxLV motif is conserved in the Vpu of subtype B of group M HIV-1, to which both NL4-3 and HV1S1 strains of HIV-1 used in this study belong. However, in the Vpu sequence from the subtype C of the group M HIV-1, the motif is replaced by the consensus sequence ExxxMV, which might or might not be functionally equivalent to the ExxxLV sequence. Interestingly, while subtype B Vpu localizes predominantly within internal membrane compartments, subtype C Vpu is substantially expressed at the cell surface (*Pacyniak et al., 2005*). In addition to the sequence variation in the ELV region, subtype C Vpu contains another putative acidic dileucine motif in the membrane-proximal region of its CD, mutation of which affected the trafficking of Vpu (*Ruiz et al., 2008*). Conceivably, subtype C Vpu hijacks AP complexes using residues distinct or partly overlapping with those used by subtype B Vpu.

The intrinsic affinity of Vpu to AP1 and AP2 suggests that Vpu has the ability to hijack clathrin-dependent trafficking pathways to target other cellular proteins with which it interacts simultaneously. Besides BST2 and CD4, HIV-1 Vpu also modulates the surface expression of two other important immune factors, NTB-A and CD1d (*Sandberg et al., 2012*). NTB-A is a co-activator for natural killer (NK) cells, and the Vpu induced downregulation of NTB-A enables infected cells to evade lysis by NK cells (*Shah et al., 2010*). This involves the mistrafficking of NTB-A, which phenotypically resembles Vpu-mediated BST2 mistrafficking (*Bolduan et al., 2013*). Although degradation is not induced, Vpu causes retention of NTB-A in a perinuclear compartment. Like BST2, the downregulation depends on an interaction between the transmembrane domains of Vpu and NTB-A (*Shah et al., 2010*). In addition, residues other than the conserved phosphorylated serines in Vpu$_{CD}$ are critical (*Bolduan et al., 2013*). In the case of CD1d, an antigen-presenting molecule in dendritic cells, downregulation by Vpu occurs through interference with the recycling of CD1d in the endosomal compartments (*Moll et al., 2010*). A critical residue of CD1d, Tyr331 in a canonical sorting motif, is key for both the natural trafficking and the Vpu-mediated CD1d downregulation. Whether CD1d also interacts with Vpu in the transmembrane or other regions remains to be determined. Given the similarities between these systems, we speculate that the modulation of NTB-A and/or CD1d by Vpu may also involve clathrin-dependent trafficking pathways.

Finally, we consider the question of why BST2 interacts on its own with AP1, since this must relate in some way to the protein's natural function. One possibility is that the interaction with AP1 allows BST2 that is internalized from the cell surface to recycle to the plasma membrane, where it acts to trap nascent virions. Another possibility derives from the ability of BST2 to induce the activity of the NF-κB family of transcription factors (*Cocka and Bates, 2012*; *Galao et al., 2012*; *Tokarev et al., 2013*); this signaling function could occur from an endosomal compartment that BST2 reaches via AP1-dependent trafficking. Yet another possibility is that BST2 uses AP1 to divert tethered virions to late endosomal compartments in antigen presenting cells, thus facilitating the presentation of viral antigens in the context of class II or class I MHC (*Neil et al., 2006*; *Miyakawa et al., 2009*; *Moffat et al., 2013*). Whether these or other processes underlie the physiologically relevant function of the interaction between BST2 and AP1 remains to be determined.

In conclusion, our biochemical and biological data support, and our crystal structure elucidates in atomic detail, how HIV-1 Vpu hijacks the AP1-dependent membrane trafficking pathway to antagonize the host restriction factor BST2. These results not only help unravel the perplexing

mechanism of Vpu-mediated antagonism of BST2 but also suggest a potentially evolutionarily conserved function of Vpu in antagonizing other host immune targets. This function might be critical in understanding the full capacity of Vpu in promoting the infectivity and pathogenesis of the primate lentiviruses.

## Materials and methods

### Cloning, expression and purification of proteins used for in vitro studies

The $BST2_{CD}$-$Vpu_{CD}$ fusion consisted of BST2 (1-21) and Vpu (HV1S1, 28-80), connected by a linker peptide of 10 amino acids: GSDEASEGSG. The encoding gene was cloned into the pMAT9s expression vector containing an N-terminal 6xHis tag followed by the maltose binding protein (MBP) and a SARS-CoV $M^{pro}$ cleavage site (*Xue et al., 2007*, *2008*). To introduce phosphorylation on Vpu S52 and S56, this plasmid and a pCDFDuet vector encoding both the α and β subunits of the casein kinase II (CK2) were co-transformed into the *E. coli* NiCo21(DE3) competent cells (New England Biolabs) for expression in the Terrific broth. Cells were induced with 0.1 mM isopropyl β-d-thiogalactopyranoside (IPTG) at $OD_{600}$ of 0.8 and grown at 16°C overnight. The protein was first purified with the Ni-NTA affinity column. For crystallization, the 6xHis-MBP tag was cleaved off by the SARS-CoV $M^{pro}$ protease. Protein was subsequently purified on a HiTrap Q anion exchange column and a Superdex 75 size exclusion column, which yielded homogenous monomeric protein. Double phosphorylation of the protein was confirmed by using MALDI mass spectrometry. For GST pull-down experiments, the 6xHis-MBP-tagged proteins, either wild-type or mutants, were purified through Ni-NTA, HiTrap Q and Superdex 200 size exclusion columns. The unphosphorylated form or the S52/56N (Vpu) mutant of the fusion was produced similarly except that the pCDFDuet-CK2 plasmid was not included during expression. $6xHis$-MBP-$Vpu_{CD}$ (phosphorylated) and $6xHis$-MBP-$BST2_{CD}$ were expressed and purified similarly.

For the AP1 core used in the crystallization, the genes encoding residues 1-613 of mouse γ and 1-158 of human σ1 were subcloned into the pCDFDuet vector, while genes encoding residues 1-584 of human β1 and 1-423 of mouse μ1 were subcloned into the pETDuet vector. Non-cleavable 6xHis tags were included at the N-termini of both the γ and β1 subunits. The heterotetrameric AP1 core was expressed overnight at 22°C in the NiCo21(DE3) cells in Terrific broth after induction by 0.1 mM IPTG. The complex was purified sequentially through Ni gravity, HiTrap Q, and Superdex 200 size exclusion columns. The μ1-CTD-truncated AP1 core was created by introducing a stop codon after residue 145 of the μ1 subunit in the pCDFDuet vector carrying both the γ and μ1 subunits. The truncated AP1 core was expressed and purified similarly. The AP1 core used in the GST pull-down experiments further included a GST tag at the C-terminus of the γ subunit. The AP1-GST complex was expressed similarly as above. The complex was purified by Ni gravity and GST columns, followed by buffer-exchange to remove the glutathione for its subsequent use in the GST pull-down assays.

MBP-μ1-CTD and MBP-μ2-CTD were created previously (*Jia et al., 2012*). MBP-μ3-CTD was created by subcloning the gene encoding residues 166-418 of rat μ3 into the pMAT9s expression vector. These proteins were overexpressed and purified as described previously (*Jia et al., 2012*). The hemicomplex of AP2, α398-σ, was created by subcloning the rat α (1-398, with three surface mutations I370A:I374S:L393A) and human σ into the pCDFDuet vector. The heterodimer was expressed similarly as the $BST2_{CD}$-$Vpu_{CD}$ fusion above and was purified sequentially on a Ni-NTA affinity column, a HiTrap S cation exchange column, and a Superdex 200 size exclusion column.

### Crystallization and data collection

Crystallization was carried out using the microbatch under-oil method. The purified AP1 core and the $BST2_{CD}$-$Vpu_{CD}$ fusion were mixed at 1:3 molar ratio to a final concentration of 4.5 mg/ml (25 mM Tris, pH 8.0, 100 mM NaCl, 0.1 mM TCEP, 0.1 mM PMSF, 0.2 mM EDTA). Equal volumes of the protein solution and the precipitant solution (100 mM Tris, pH 7.0, 150 mM NaCl, 8% PEG6000) were mixed. The drop was sealed using a mixture of paraffin and silicon oil at a 2:1 ratio. Crystals appeared within 24 hr at room temperature and grew to full size in about a week.

Crystals were cryo-protected using the precipitant solution containing 20% glycerol and then frozen in liquid nitrogen. Datasets were collected at NE-CAT beamline 24ID-C at the Advanced Photon

Source, Argonne National Laboratory, and beamline X29 at the National Synchrotron Light Source, Brookhaven National Laboratory. The crystals were in the P43 space group and diffracted to a highest resolution of 3.0 Å. The data collection statistics are summarized in *Table 1*.

## Structure determination and refinement

The structural solution was obtained by molecular replacement using PHASER (*McCoy et al., 2007*) implemented in PHENIX (*Adams et al., 2010*). Only one molecule exists in the asymmetric unit. The structure of the closed AP1 core (PDB ID: 1W63) (*Heldwein et al., 2004*) was divided into three search models: the γ and σ1 hemicomplex, the β1 and μ1-NTD hemicomplex, and the μ1-CTD. These models were used in sequential searches to successfully obtain the solution. Iterative rounds of model building in COOT (*Emsley et al., 2010*) and refinement with PHENIX (*Adams et al., 2010*) were carried out. Data sharpening was performed to enhance the details in the electron density map (*Liu and Xiong, 2013*). $BST2_{CD}$ and $Vpu_{CD}$ were built into the model based on the prominent difference densities at their corresponding locations. The final model has an $R_{work}/R_{free}$ of 0.18/0.22. The refinement statistics are summarized in *Table 1*.

## In vitro binding assays using size exclusion chromatography

The purified proteins, MBP-μ1/2/3-CTD (0.86 mg) or α398-σ (0.73 mg) and $MBP-BST2_{CD}/MBP-Vpu_{CD}$ (0.73 mg) or their mutants, were mixed to a final volume of 500 μl. Similarly, the μ1-CTD-truncated AP1 (3.4 mg) and $MBP-Vpu_{CD}$ (1.2 mg, wt or the ELV/AAA mutant) were mixed to a final volume of 500 μl. All samples were incubated overnight at 4°C and then applied to the Superdex 200 10/300 GL size exclusion column pre-equilibrated with the elution buffer (25 mM Tris, pH 8, 100 mM NaCl, 0.1 mM TCEP). The elution fractions were analyzed by using SDS PAGE.

## In vitro GST pulldown assays

The purified proteins, AP1-GST (0.24 mg) and $MBP-BST2_{CD}-Vpu_{CD}/MBP-BST2_{CD}/MBP-Vpu_{CD}$ (0.18 mg) or their mutants, were mixed in a final volume of 100 μl and incubated at 4°C overnight. The protein solution was then loaded onto a small gravity flow column containing 0.2 ml GST resin. Flow through was collected and the resin was extensively washed with 5 × 0.9 ml GST binding buffer (50 mM Tris, pH 8, 100 mM NaCl, 0.1 mM TCEP). The bound proteins were then eluted with 5 × 0.1 ml GST elution buffer containing 10 mM reduced glutathione. The eluted proteins were analyzed on SDS PAGE stained with Coomassie blue.

## In vitro 6xHis pulldown assays

6xHis-MBP-μ1 (0.25 mg) and the $BST2_{CD}-Vpu_{CD}$ fusion (0.1 mg) or its mutants were mixed in a final volume of 100 μl and incubated at 4°C for 2 hr. The proteins were loaded on a small gravity column containing 0.2 ml Ni-NTA resin. Flow through was collected and the resin was extensively washed with 5 × 0.9 ml binding buffer containing 20 mM imidazole. The bound proteins were subsequently eluted with 5 × 0.1 ml Ni elution buffer containing 400 mM imidazole. The eluted proteins were analyzed on SDS PAGE stained with Coomassie blue.

## Yeast two-hybrid assays

AH109 yeast cells (Clontech Laboratories, Inc, Palo Alto, CA) were co-transformed to express both hybrid

**Table 1.** Crystallographic data collection and refinement statistics

| | Native |
|---|---|
| Data collection | |
| Space group | $P4_3$ |
| Cell dimensions | |
| *a*, *b*, *c* (Å) | 160.5, 160.5, 118.4 |
| Wavelength (Å) | 0.9792 |
| Resolution (Å) | 48.8–3.0 (3.11–3.0) |
| $R_{merge}$ | 0.077 |
| $I/\sigma I$ | 13.6 (0.9) |
| Completeness (%) | 99.4 (99.5) |
| Redundancy | 3.8 (3.7) |
| Refinement | |
| Unique reflections | 59937 |
| $R_{work}/R_{free}$ | 0.186/0.229 |
| No. atoms | |
| Protein | 13,926 |
| Water | 14 |
| *B*-factors | |
| Protein | 107.1 |
| Water | 74.2 |
| R.m.s deviations | |
| Bond lengths (Å) | 0.013 |
| Bond angles (°) | 1.27 |
| Ramachandran | |
| Favored | 95% |
| Outliers | 0.23% |

Values in parenthesis are for highest-resolution shell.

proteins. The GAL4-Activation domain in pACT2 fused with the μ1 subunit was provided by Juan Bonifacino. The GAL4-DNA binding domain (DBD) in pGBT9 was fused to the BST2 cytoplasmic domain (CD) with the linker GGGSGGGSGGGS inserted between the BST2$_{CD}$ start codon and the DBD of the GAL4 protein. BST2$_{CD}$ was inserted between the *Eco*RI and *Sal*I restriction sites of the pGBT9 multiple cloning site. Yeast cells were transformed by the lithium-acetate method. Transformants were picked; colonies were pooled and grown in -Leu/-Trp liquid media before plating as spots on -Leu/-Trp solid media as a growth control or on -Trp/-Leu/-His solid media containing 3-aminotriazole (3AT) to test for interaction.

## Virion release assays

For the virion release assays, the following plasmids were used: the proviral Vpu-mutant pNL43/Udel (*Klimkait et al., 1990*); pcDNA3.1-BST-2 from Autumn Ruiz and Edward Stephens; pcDNA3.1-BST2-Y6/8A constructed by site-directed mutagenesis of pcDNA3.1-BST2 using the QuikChange II Site-Directed Mutagenesis Kit (Stratagene, La Jolla, CA); pVpHu-FLAG constructed by cloning the codon-optimized Vpu sequence from pVpHu (from Klaus Strebel) with primers introducing a C-terminal FLAG tag (DYKDDDDK) into the *Nhe*I and *Xho*I sites of the pcDNA3.1(−) vector (Life Technologies, Carlsbad, CA); and the FLAG-tagged Vpu-mutant expression plasmids pVpHu-FLAG-S52/56N, pVpHu-FLAG-ELV59,63,64AAA and pVpHu-FLAG-S52/56N + ELV/AAA constructed by site-directed mutagenesis of pVpHu-FLAG using QuikChange (Stratagene).

HEK293T cells in six-well plates were transfected using Lipofectamine 2000 (Life Technologies, Carlsbad, CA) with 50 ng of the indicated BST2 expression plasmid, 250 ng of the indicated VpHu-FLAG expression plasmid and 3600 ng of the Vpu-mutant proviral plasmid pNL43/Udel. Supernates were collected 24 hr after transfection as previously described (*Van Damme et al., 2008*). Virion-associated p24 was pelleted through a 20% sucrose cushion before measurement by p24 ELISA (Advanced Bioscience Laboratories, Rockville, MD). Cells were harvested and lysed 24 hr post-transfection and immunoblots for BST2, gp120, actin, and FLAG-epitope were performed as previously described (*Day et al., 2004*; *Tokarev and Guatelli, 2011*; *Tokarev et al., 2013*).

## Co-immunoprecipitation assays

293T cells were transfected with the plasmids expressing the constructs indicated in the figure legends, using Lipofectamine 2000 and the manufacturer's protocol. The next day, cells were lysed in lysis buffer (50 mM Tris HCl pH7.4, 150 mM NaCl, 1 mM EDTA, 1% Triton X-100 and 5% glycerol) supplemented with protease inhibitors cocktail (Roche Diagnostics). Lysates were cleared by centrifugation for 10 min at 16,000×*g* and incubated with magnetic anti-FLAG-coated beads for BST2, Vpu or the BST2-Vpu chimera; or with anti-HA-coated beads for β-TrCP, which were pre-blocked with 3% BSA in PBS, for 2 hr at 4°C with continuous rotation. Beads were washed three times with lysis buffer containing 250 mM NaCl. The precipitated material was eluted with boiling in 2x Laemmli buffer and subjected to Western blotting. Endogenous μ1 was detected using a rabbit antiserum provided by Linton Traub.

## Immunofluorescence assays

HeLa P4R5 or HT1080 cells were transfected with Vpu or BST2-Vpu chimera constructs, as indicated in the figure legends, using Lipofectamine 2000. The next day, cells were fixed with 4% para-formaldehyde, permeabilized with 0.2% NP-40, blocked with PBS containing 5% donkey serum, 5% goat serum (Jackson Immunoresearch) and 3% BSA. For Figure S6, cells were first stained with mouse anti-γ adaptin (clone 100/3, Sigma-Aldrich) and anti-Vpu rabbit serum, washed three times with PBS and stained with anti-rabbit antibody conjugated to rhodamine red-X and anti-mouse antibody conjugated to FITC (Jackson Immunoresearch). Cells were then blocked with 5% mouse serum and stained with mouse anti-BST2 conjugated to Alexa Fluor 647 (Biolegend, San Diego CA). For *Figure 9A*, the staining was performed as just described, except that the secondary antibody to detect Vpu was anti-rabbit conjugated to FITC and the secondary antibody to detect AP1 was anti-mouse conjugated to rhodamine red-X. For *Figure 9C*, the chimeras were visualized using anti-Vpu rabbit serum, and AP1 was visualized using mouse anti-γ adaptin. The secondary antibodies used were anti-mouse conjugated to FITC and anti-rabbit conjugated to rhodamine red-X. After staining, cells were washed, fixed again, and mounted in solid media (Fluka). Images were acquired as Z-series using an Olympus microscope in wide-field mode. Images were deconvolved using a nearest neighbors method (SlideBook software, Intelligent Imaging Innovations, Denver,

CO) and the Z-series were collapsed into single projection images. Co-localization was quantified using the Pearson correlation coefficient function in SlideBook. T-values were calculated using Microsoft Excel. Composite images were generated using Adobe Photoshop.

## Flow cytometry

Two-color flow cytometry was performed on unpermeabilized cells using an Accuri C6 flow cytometer to measure the intensity of GFP and the FLAG-epitope, which was detected indirectly using murine anti-FLAG (Sigma-Aldrich) followed by allophycocyanin (APC)-conjugated anti-mouse antibody (Biolegend, San Diego, CA).

## Acknowledgements

We thank Mark Skasko for the initial design and functional characterization of the RL/AA mutant. We thank the staff at the Advanced Photon Source beamline 24-ID and the National Synchrotron Light Source beamline X29. We thank Juan Bonifacino for providing the genes of AP complexes, Linton Traub for providing the rabbit antiserum to μ1, and Olaf-Georg Issinger for providing the CKII genes. We thank Suresh Subramani for providing the AH109 yeast cells. Rabbit antisera to BST2 and Vpu were obtained from the NIH AIDS Reagent Program and contributed by Klaus Strebel. This work was supported by US National Institutes of Health (NIH) grants AI102778 (YX and JG), AI097064 (YX), AI081668 (JG) as well as by The James B Pendleton Charitable Trust. MR was supported in part by the San Diego Fellowship. AT was supported in part by AI007384 (T32 AIDS Training Grant).

## Additional information

### Funding

| Funder | Grant reference number | Author |
|---|---|---|
| National Institutes of Health | AI102778 | John Guatelli, Yong Xiong |
| The James B. Pendleton Charitable Trust | | John Guatelli |
| The San Diego Fellowship | | Maryan Rizk |
| National Institutes of Health | AI097064 | Yong Xiong |
| National Institutes of Health | AI081668 | John Guatelli |
| National Institutes of Health | AI007384 | Andrey Tokarev |

The funders had no role in study design, data collection and interpretation, or the decision to submit the work for publication.

### Author contributions

XJ, EW, AT, ML, MR, MS, Conception and design, Acquisition of data, Analysis and interpretation of data, Drafting or revising the article; JG, YX, Conception and design, Analysis and interpretation of data, Drafting or revising the article

## Additional files

### Major datasets

The following datasets were generated:

| Author(s) | Year | Dataset title | Dataset ID and/or URL | Database, license, and accessibility information |
|---|---|---|---|---|
| Jia X, Xiong Y | 2014 | Crystal structure of the human BST2 cytoplasmic domain and the HIV-1 Vpu cytoplasmic domain bound to the clathrin adaptor protein complex 1 (AP1) core | http://www.rcsb.org/pdb/explore/explore.do?structureId=4P6Z | Publicly available at the RCSB Protein Data Bank (http://www.rcsb.org/). |

The following previously published datasets were used:

| Author(s) | Year | Dataset title | Dataset ID and/or URL | Database, license, and accessibility information |
|---|---|---|---|---|
| Heldwein E, Macia E, Wang J, Yin HL, Kirchhausen T, Harrison SC | 2004 | AP1 clathrin adaptor core | http://www.rcsb.org/pdb/explore/explore.do?structureId=1W63 | Publicly available at the RCSB Protein Data Bank (http://www.rcsb.org/). |
| Owen DJ, Mccoy AJ, Kelly BT, Evans PR | 2008 | AP2 clathrin adaptor corewith CD4 Dileucine peptide RM(phosphoS) EIKRLLSE Q to E mutant | http://www.rcsb.org/pdb/explore/explore.do?structureId=2jkt | Publicly available at the RCSB Protein Data Bank (http://www.rcsb.org/). |
| Ren X, Farias GG, Canagarajah BJ, Bonifacino JS, Hurley JH | 2013 | Structural basis for recruitment and activation of the AP-1 clathrin adaptor complex by Arf1 | http://www.rcsb.org/pdb/explore/explore.do?structureId=4hmy | Publicly available at the RCSB Protein Data Bank (http://www.rcsb.org/). |
| Owen DJ, Evans PR | 1998 | Mu2 adaptin subunit (AP50) of AP2 adaptor (second domain), complexed with TGN38 internalization peptide dyqrln | http://www.rcsb.org/pdb/explore/explore.do?structureId=1bxx | Publicly available at the RCSB Protein Data Bank (http://www.rcsb.org/). |
| Mardones GA, Kloer DP, Burgos PV, Bonifacino JS, Hurley JH | 2012 | Crystal structure of adaptor protein complex 3 (AP-3) mu3A subunit C-terminal domain, in complex with a sorting peptide from TGN38 | http://www.rcsb.org/pdb/explore/explore.do?structureId=4ikn | Publicly available at the RCSB Protein Data Bank (http://www.rcsb.org/). |
| Willbold D, Hoffmann S, Rosch P | 1997 | NMR solution structure of the HIV-1 Vpu cytoplasmic domain, 9 structures | http://www.rcsb.org/pdb/explore/explore.do?structureId=1vpu | Publicly available at the RCSB Protein Data Bank (http://www.rcsb.org/). |
| Wittlich M, Koenig BW, Willbold D | 2008 | Solution fold of HIV-1 Virus protein U cytoplasmic domain in the presence of DPC micelles | http://www.rcsb.org/pdb/explore/explore.do?structureId=2k7y | Publicly available at the RCSB Protein Data Bank (http://www.rcsb.org/). |

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
