## [Decision Letter]

Thank you for sending your work entitled “Structural basis of HIV-1 Vpu-mediated BST2 antagonism via hijacking of the clathrin adaptor protein complex 1” for consideration at *eLife*. Your article has been favorably evaluated by a Senior editor (John Kuriyan), a Reviewing editor (Wes Sundquist), and 2 reviewers.

The Reviewing editor and the reviewers discussed their comments before we reached this decision, and the Reviewing editor has assembled the following comments to help you prepare a revised submission.

The authors report the crystal structure of the core of the AP1 clathrin adaptor complex (AP1) in complex with a designed fusion construct that comprises the HIV-1 Vpu and BST2 cytoplasmic domains. The structure reveals that both Vpu and BST2 bind AP1. BST1 binds the tyrosine binding site on the μ1 site of AP1 via an unusual YxYxxV interaction that seems to explain the observed preference for μ1 (vs. μ2 or μ3). Vpu uses an “ELV” motif to bind the γ and σ1 subunits of AP1 through a canonical acidic dileucine-type interaction. The R44 and L/I45 residues of Vpu also appear to make an additional contact adjacent to a novel AP1 interface formed between the μ1 and γ subunits, although this interaction is not well defined by the electron density. The new μ1-γ interface can form because the AP1 core adopts a more open conformation that has been seen in previous structures. Thus, Vpu provides additional interactions that enable the viral protein to augment an intrinsic interaction between BST2 and AP1, and the interactions apparently function together to help retain BST2 in juxtanuclear endosomes and to stimulate lysosomal degradation. Qualitative in vitro binding and mutational analyses support the importance of the observed interactions for Vpu-BST2-AP1 complex formation in vitro. Presented experiments also support the relevance of these interactions for formation of functional BST2-Vpu-AP1 complexes in cells, although the effects are not dramatic in all cases (e.g., Figure 9), and the analyses are complicated by the fact that Vpu appears to downregulate BST2 through multiple different mechanisms.

Overall, the structural studies are technically strong and represent an elegant use of crystallography to establish an unexpected mechanism whereby HIV Vpu induces BST2 mistrafficking by hijacking the cellular AP1 complex. Thus, the work reveals one molecular mechanism of HIV-1 Vpu-induced downregulation of BST2 and is an important advance in our understanding of a key component in the battle between HIV and the innate immune system.

The most important outstanding issue is the relative importance of AP1 in Vpu function. The studies presented here support and extend previous observations that: 1) the acidic-dileucine motif in Vpu is essential for trafficking of BST2 for endosomal degradation, 2) this signal works additively with an endocytic recycling motif in BST2 itself, and 3) a role for AP1 is likely, but is difficult to prove definitively owing to potential redundancy/compensation in clathrin transport pathways and to other mechanisms of Vpu action. The studies presented here (and previously), however, fall short of unequivocally demonstrating the role of AP1 in Vpu function. Quantitative analyses of the different Vpu-BST2-AP1 interactions are also lacking, as are experiments that establish a definitive role for the novel interaction between the Vpu(R44/I45) motif and AP1. The current studies are already very strong and informative and the missing experiments may be technically challenging so we have divided them into issues that must be addressed prior to publication vs. issues that would significantly enhance the impact of the paper if they can be addressed.

*Issue that must be addressed prior to publication*:

The virological and cellular phenotypes of the R44/I45 mutation should be tested in light of the newly observed μ1 interaction. The referenced paper that is currently used to justify the phenotypic importance of these residues is fairly rudimentary and this is unsatisfactory given that the Vpu-μ1 interaction interaction is one of the important new findings in the current study. The authors may also want to note that a very recent publication by Pickering et al identified naturally occurring mutants in these residues has having impaired function.

*Issues that the authors should seriously consider addressing prior to publication*:

1) Efficient AP1 knockdown is challenging, but the molecular interactions between Vpu, BST2 and AP1 are have now been identified. Single point mutations in these interactions – V64 in Vpu for example, or the R44, I45 mutations – might be expected to weaken (but not abolish) the AP1 interaction. It may therefore be possible to reveal a “synthetic lethal” AP1 knockdown phenotype for BST2 counteraction under these conditions. This would add considerably to the manuscript.

2) Along a similar line, BST2 degradation is easily measured in infected cells, but this has not yet been tested under AP1 knockdown conditions. This phenotype may be more amenable to study than antagonism (because partial knockdowns may have a magnified effect on final BST2 degradation, rather than on antagonism, where ultimate traffic to late endosomes is probably not required).

3) Studies of the type being presented usually report ITC, SPR, or other quantitative binding data. Given the well-behaved reagents in hand and the skills of the authors, they should consider quantitating the affinities of the following interactions: Vpu tail with AP1 (Δ-μ1 CTD) and full AP1 core; BST2 tail with μ1 CTD and full AP1 core; and Vpu-BST fusion with the full AP1 core.

*Other points*:

1) Figure 9. The different panels showing raw flow cytometry data are not visually informative. Consider showing just the quantified data (i.e., the far right panels).

2) The authors should discuss possible functions of the AP1-BST2 interaction in uninfected cells (i.e., in the absence of Vpu).

3) Figure 5 and text: It is difficult to discern differences between the overall domain orientations and those seen in the Jackson and Ren structures, except for the new μ1-γ interface. It would be helpful to have a more quantitative comparison of the subunit orientations, say in terms of Euler angles relating the domain positions to one another.

4) Given that the ELV motif of Vpu binds canonically to the sigma pocket, can the authors definitively rule out possible interactions between the BST2(CD)-Vpu(CD) fusion construct and AP2? Any evidence that Vpu ELV can occupy the sigma site in AP2 could imply a level of redundancy that would explain the difficulty in ascribing a phenotype to the AP1 knockdown alone, and would provide a plausible explanation for the observation that exclusion of budding virions at the plasma membrane is dependent on helix 2 (of particular relevance because some of the co-authors have previously described a partial phenotype for an AP2 knockdown).

5) In HIV-1 clades C and F, the acidic dileucine in Vpu is positioned close to the membrane in helix 1. Can the authors speculate on possible mechanistic differences given that clade C Vpu's have been reported to localize differently?

6) Both Dube et al and Vigan and Neil should also be referenced for the determinants of BST2/Vpu interactions (in addition to the references that are cited).

---

## [Author Response]

Issue that must be addressed prior to publication:

*The virological and cellular phenotypes of the R44/I45 mutation should be tested in light of the newly observed μ1 interaction. The referenced paper that is currently used to justify the phenotypic importance of these residues is fairly rudimentary and this is unsatisfactory given that the Vpu-μ1 interaction interaction is one of the important new findings in the current study. The authors may also want to note that a very recent publication by Pickering et al identified naturally occurring mutants in these residues has having impaired function*.

We agree with the need for virological and cellular validation of the R44/I45 mutations, which would support our structural and biochemical observations of the potential involvement of R44 and I45 in binding to AP1 (Figure 7). We have now included such data in the results section (Figure 7). We indeed observed that the RL/AA mutation impaired the abilities of Vpu to downregulate the surface level of BST2 (Figure 7) and to promote virion release (Figure 7). The RL/AA mutation has an additive effect on top of the S52/56N mutation on virion release (Figure 7), supporting the hypothesis that the motifs affect separate pathways for antagonizing BST2. We also thank the reviewers for directing us to the work by Pickering et al, where the phenotypes from naturally occurring mutations on L/I45 support our findings. We have now included this reference in the manuscript.

Issues that the authors should seriously consider addressing prior to publication:

*1) Efficient AP1 knockdown is challenging, but the molecular interactions between Vpu, BST2 and AP1 are have now been identified. Single point mutations in these interactions – V64 in Vpu for example, or the R44, I45 mutations – might be expected to weaken (but not abolish) the AP1 interaction. It may therefore be possible to reveal a “synthetic lethal” AP1 knockdown phenotype for BST2 counteraction under these conditions. This would add considerably to the manuscript*.

We agree that it could be a good idea to combine mutations in Vpu that collectively and specifically abolish its binding with AP1 and look at the phenotypes for the “synthetic” AP1 knockdown. However, as Vpu interacts with both AP1 and AP2, it is possible that such Vpu mutations may also interfere with the binding between Vpu and AP2 and associated trafficking pathways. We would very much like to test these ideas. However, due to the complex nature of the experiments and the time constraints for resubmission, we are not able to address these questions in the detail that they deserve for this manuscript.

*2) Along a similar line, BST2 degradation is easily measured in infected cells, but this has not yet been tested under AP1 knockdown conditions. This phenotype may be more amenable to study than antagonism (because partial knockdowns may have a magnified effect on final BST2 degradation, rather than on antagonism, where ultimate traffic to late endosomes is probably not required)*.

We thank the reviewers for the suggestion and agree that looking at the BST2 degradation phenotype under AP1 knockdown conditions could lead to new insights in the antagonism. However, we are not able to test this in a timely manner before resubmission.

*3) Studies of the type being presented usually report ITC, SPR, or other quantitative binding data. Given the well-behaved reagents in hand and the skills of the authors, they should consider quantitating the affinities of the following interactions: Vpu tail with AP1 (Δ-μ1 CTD) and full AP1 core; BST2 tail with μ1 CTD and full AP1 core; and Vpu-BST fusion with the full AP1 core*.

These are excellent suggestions. We devoted substantial effort to the quantification of binding affinities of the interactions using ITC. However, due to the relative low interaction affinity, multiple steps involved in the binding process (conformational changes in AP1 and subsequent binding), and the poor solution behavior of the interactions components (such as μ1 CTD), our repeated attempts have not yielded reliable data.

Other points:

*1)*
Figure 9*. The different panels showing raw flow cytometry data are not visually informative. Consider showing just the quantified data (i.e., the far right panels)*.

We have made the change as suggested.

*2) The authors should discuss possible functions of the AP1-BST2 interaction in uninfected cells (i.e., in the absence of Vpu)*.

We have added discussion of the AP1-BST2 interaction in the context of its functions in the cell in the results section where BST_CD_ binding with different μ subunits are presented. We have also added a paragraph to the Discussion section that is devoted to the possible reasons that BST2 traffics via AP1 in the context of known and hypothetical functions of the protein.

*3)*
Figure 5
*and text: It is difficult to discern differences between the overall domain orientations and those seen in the Jackson and Ren structures, except for the new μ1-γ interface. It would be helpful to have a more quantitative comparison of the subunit orientations, say in terms of Euler angles relating the domain positions to one another*.

We have added the relative rotation of γ/σ1 in the current AP1 from that in the earlier structures. In addition, we have changed the representation in the figure overlaying the structures (Figure 6) so that the structure by Ren et al. (surface representation) can be clearly distinguished from the current structure (ribbon representation). We believe the figure is substantially improved to show the differences in domain orientations.

*4) Given that the ELV motif of Vpu binds canonically to the sigma pocket, can the authors definitively rule out possible interactions between the BST2(CD)-Vpu(CD) fusion construct and AP2? Any evidence that Vpu ELV can occupy the sigma site in AP2 could imply a level of redundancy that would explain the difficulty in ascribing a phenotype to the AP1 knockdown alone, and would provide a plausible explanation for the observation that exclusion of budding virions at the plasma membrane is dependent on helix 2 (of particular relevance because some of the co-authors have previously described a partial phenotype for an AP2 knockdown)*.

We thank the reviewers for the excellent suggestion. We have now added new data showing that Vpu also interacts with AP2 subunits (α398-σ and μ2-CTD), but not with μ3-CTD of AP3 (Figure 2). This points to the ability of Vpu to hijack multiple AP complexes, adding to the argument that redundant usage of AP complexes might contribute to the complexity of the BST2 antagonism by Vpu.

*5) In HIV-1 clades C and F, the acidic dileucine in Vpu is positioned close to the membrane in helix 1. Can the authors speculate on possible mechanistic differences given that clade C Vpu's have been reported to localize differently*?

Indeed the ELV motif is not well conserved in HIV-1 clade/subtype C, and another putative acidic dileucine motif is found in the juxtamembrane region of the cytoplasmic domain of the clade C Vpu. It has been reported that mutation of this motif affected the cellular localization of Vpu. As suggested, we have now discussed that subtype C Vpu may hijack AP complexes using residues distinct or partly overlapping with those used by subtype B Vpu.

*6) Both Dube et al and Vigan and Neil should also be referenced for the determinants of BST2/Vpu interactions (in addition to the references that are cited)*.

We thank the reviewers for directing us to the two references. We have added them to the Introduction where we discussed the determinants of BST2/Vpu interactions.